



# An explicit GPU-based material point method solver for elastoplastic problems (ep2-3De v1.0)

Emmanuel Wyser[1,2], Yury Alkhimenkov[1,2,3], Michel Jaboyedoff[1,2], and Yury Y. Podladchikov[1,2,3]

[1]Institute of Earth Sciences, University of Lausanne, 1015 Lausanne, Switzerland
[2]Swiss Geocomputing Center, University of Lausanne, 1015 Lausanne, Switzerland
[3]Faculty of Mechanics and Mathematics, Lomonosov Moscow State University, Moscow, 119991, Russia

**Correspondence:** Emmanuel Wyser (manuwyser@gmail.com)

**Abstract.**

We propose an explicit GPU-based solver within the material point method (MPM) framework on a single graphics processing unit (GPU) to resolve elastoplastic problems under two- and three-dimensional configurations (i.e., granular collapses and slumping mechanics). Modern GPU architectures, including Ampere, Turing and Volta, provide a computational framework that is well suited to the locality of the material point method in view of high-performance computing. For intense and nonlocal computational aspects (i.e., the back-and-forth mapping between the nodes of the background mesh and the material points), we use straightforward atomic operations (the scattering paradigm). We select the generalized interpolation material point method (GIMPM) to resolve the cell-crossing error, which typically arises in the original MPM, because of the $C_0$ continuity of the linear basis function. We validate our GPU-based in-house solver by comparing numerical results for granular collapses with the available experimental data sets. Good agreement is found between the numerical results and experimental results for the free surface and failure surface. We further evaluate the performance of our GPU-based implementation for the three-dimensional elastoplastic slumping mechanics problem. We report i) a maximum performance gain of x200 between a CPU- and GPU-based implementation, provided that ii) the hardware limit (i.e., the peak memory bandwidth) of the device is reached. We finally showcase an application to slumping mechanics and demonstrate the importance of a three-dimensional configuration coupled with heterogeneous properties to resolve complex material behaviour.

## 1 Introduction

Graphics processing units, or GPUs, have revolutionized the entire field of high-performance computing (HPC) in the last decade. GPUs are many-core processors that were originally developed by the gaming industry in the mid-1990s to accelerate graphics and video rendering. Currently, GPUs are widely employed hardware accelerators used in various applications, including artificial intelligence (AI) and machine learning. GPUs are also increasingly used for high-performance scientific computing (see Dong et al. 2015b; Omlin et al. 2018; Räss et al. 2018; Zhang et al. 2021; Alkhimenkov et al. 2021). The majority of the scientific algorithms on many-core (e.g., GPU) hardware accelerators are memory-bounded, meaning that data transferring (reading and writing) limits the performance of a solver. This is in contrast to the recent CPU-bounded algorithms, where arithmetic floating point calculations are the main limiting factor in solver performance. This GPU supercomputing



breakthrough requires re-engineering existing scientific codes or developing new algorithmic structures to efficiently take advantage of the intrinsic low-level parallelism of GPUs.

The material point method (MPM) was first proposed by Sulsky et al. (1994) and was further advanced by the generalized interpolation material point method (GIMPM) by Bardenhagen and Kober (2004). It can be think of as a finite element method (FEM) in which a) integration points (i.e., material points) move and b) convey state variables, e.g., stress and strain

components. The continuum is discretized by material points. The nodal momentum equations are solved on a background mesh and nodal basis functions provide a mapping framework between the mesh and the material points to transfer either the updated nodal solution or material point properties. The background mesh can be reset and actually never deforms. It has been widely used for large deformation geomechanical problems such as retrogressive failure, coupled hydromechanical landslides or granular collapses (Tran and Sołowski, 2019; Bandara and Soga, 2015; Dunatunga and Kamrin, 2015).

From a computational point a view, it is critical that the MPM be able to simulate large-scale problems in both two- and three-dimensional configurations. From this perspective, a few researchers have exploited parallel computing using a single or multiple GPU strategy (Dong et al., 2015a; Dong and Grabe, 2018) to efficiently implement an explicit GIMPM for two-dimensional configurations. More recently, some researchers in the graphics community presented a similar implementation (Gao et al., 2018; Hu et al., 2019; Wang et al., 2020) for three-dimensional configurations. One of the most computationally

expensive operations in MPM is mapping between material points and their associated nodes, which is supported by basis functions. When implementing a GPU, the two most common approaches are *gathering* and *scattering*. The former gathers the material point's state variables (i.e., mass, velocity component or stresses) to the nodes, whereas the latter scatters (i.e., distributes) the material point's state variables to their associated nodes. This leads to write conflicts, as several threads are writing into the same memory location at the same time. Gao et al. (2018) demonstrated the superiority of *scattering* over

*gathering*, provided that the write conflicts are handled without atomic operations. Gao et al. (2018) proposed parallel scattering that results in a performance of an order of magnitude higher than that of a naive atomic implementation. Recently, Wang et al. (2020) proposed an Array of Structures of Arrays (AoSoA) as an efficient layout. It is largely responsible for CPU or GPU performances, as it dictates the memory access pattern (Wang et al., 2020) by ensuring coalesced memory accesses.

We propose an explicit GIMPM implementation in a three-dimensional configuration on a single GPU (ep2-3De v1.0), taking

advantage of the efficient vectorized algorithmic structure of the MPM solver proposed by Wyser et al. (2020a). Our GPU-based solver relies on built-in functions of atomic operations for the mapping between material points and their associated nodes (i.e., scattering). For large-scale simulations, the main hardware limit is the GPU on-chip memory, which was well documented by Dong and Grabe (2018). The GPU solver ep2-3De v1.0[1] combines MATLAB for pre- and postprocessing activities with the massive power of the most recent GPU architectures available (Ampere, Turing and Tesla architectures). This approach allows

the user to easily set the problem's geometry and initialize the material points as well as their state variables. Everything needed is then passed to the GPU, which further performs the computations. We propose a formal framework to evaluate the

---

[1]The routines of the ep2-3De v1.0 solver are available for download from Bitbucket at: https://bitbucket.org/ewyser/ep2-3de/src/master/ (last access: June 16, 2021). The routines archive (v1.0) (Wyser et al., 2021) is available from a permanent DOI repository (Zenodo) at https://doi.org/10.5281/zenodo.4966590 (June 16, 2021).





performance of our GPU-based implementation based on the metric for memory-bounded codes, i.e., the effective memory
throughput (Omlin, 2017). Since the memory wall has been reached, the memory bandwidth becomes the limiting factor for
performance. In addition, it is an easily comparable metric. Similarly, we also report the average number of iterations per
second for the same reason: it indicates a relative performance, and it does not depend on material properties (e.g., bulk or
shear moduli). We also implement the solver ep2-3De v1.0 under a single-CPU architecture to provide a reference baseline
for the performance evaluation of the GPU-based implementation. For the validation of our solver, we simulate the granular
collapse problem in a three-dimensional configuration and compare the result against the well-known experimental results of
Bui et al. (2008).

## 2  Numerical implementation

In this section, we briefly describe the governing equations implemented in the MPM solver. We use a linear elastoplastic
rheology. Large deformations are carried out via a rate-dependent formulation with the Jaumann stress rate.

### 2.1  Governing equations

The conservation of linear momentum is given by (using the Einstein summation convention)

$$\rho \frac{\partial v_k}{\partial t} = \frac{\partial \sigma_{kl}}{\partial x_l} + \rho g_k, \tag{1}$$

where $\sigma_{kl}$ is the Cauchy stress tensor, $v_k = \partial u_k/\partial t$ is the velocity, $u_k$ is the displacement, $g_k$ is the body force, and $k,l = \overline{1..3}$.
The conservation of angular momentum is given by $\sigma_{kl} = \sigma_{lk}$. Dirichlet and Neumann boundary conditions are

$$u_k = \bar{u}_k \quad \text{on} \quad \partial \Omega_u, \tag{2}$$

$$\sigma_{kl} n_l = \bar{\tau}_k \quad \text{on} \quad \partial \Omega_\tau, \tag{3}$$

where $\hat{u}_k$ and $\hat{\tau}_k$ are prescribed displacements, and $n_k$ is a unit normal vector pointing outward from the boundary $\partial \Omega$ of the
domain $\Omega$. Following the standard FEM procedure, we use the updated Lagrangian framework; thus, the weak form of Eq. 1
is written in the current spatial configuration. The weak form of Eq. 1 can be obtained by multiplying it with a test function $\phi$
and then applying integration by parts and divergence theorem, leading to

$$\int_\Omega \phi \rho a_k \mathrm{d}\Omega = \int_\Omega \phi \rho g_k \mathrm{d}\Omega - \int_\Omega \frac{\partial \phi}{\partial x_l} \sigma_{kl} \mathrm{d}\Omega + \int_{\partial \Omega_\tau} \phi \bar{\tau}_k \mathrm{d}S, \tag{4}$$

where $\partial v_k/\partial t = a_k$ is the acceleration, $\phi$ is any test function that vanishes on $\partial \Omega_u$, and $\bar{\tau}_k$ is the external traction applied on
the boundary $\partial \Omega$, $k = \overline{1..3}$. However, in our MPM implementation, tractions on the boundary are not used. Eq. 4 can be solved
using a finite element approach leading to the following compact form:

$$[M_{ij}a_j]_k = \left[f_i^{\text{ext}} - f_i^{\text{int}}\right]_k, \tag{5}$$





where $M_{ij} = \sum_{p=1}^{n_p} m_p \phi_i(\boldsymbol{x}_p)\phi_j(\boldsymbol{x}_p)$ is the consistent mass matrix with $\phi_i(\boldsymbol{x}_p)$ being the basis function between node $i$ and

85 material point $p$. This work adopts a lumped mass matrix, i.e., $m_i \equiv M_{ii} = \sum_{p=1}^{n_p} m_p \phi_i(\boldsymbol{x}_p)$, to avoid an expensive matrix inversion (Sulsky et al., 1994; Bardenhagen and Kober, 2004; González Acosta et al., 2020). The external $f_{k,n}^{\text{ext}}$ and internal $f_{k,n}^{\text{int}}$ forces at node $n$ are then defined by

$$f_{k,n}^{\text{ext}} = \sum_{p=1}^{n_p} m_p \phi_n(\boldsymbol{x}_p)g_k, \tag{6}$$

$$f_{k,n}^{\text{int}} = \sum_{p=1}^{n_p} v_p \frac{\partial \phi_n}{\partial x_l}(\boldsymbol{x}_p)\sigma_{kl,p}, \tag{7}$$

where $m_p$ is the material point's mass, $v_p$ is the material point's volume and $\sigma_{kl,p}$ is the material point's Cauchy stress tensor. Solving Eq. 5 for the acceleration $a_{k,n}$, the updated velocity is obtained via a forward-Euler scheme,

$$v_{k,n}^{t+\Delta t} = v_{k,n}^t + \Delta t a_{k,n}, \tag{8}$$

where the velocity is given by $v_{k,n}^t = m_n^{-1} \sum_{p=1}^{n_p} \phi_n(\boldsymbol{x}_p)m_p v_{k,p}$ and $v_{k,p}$ is the material point's velocity. Boundary conditions are enforced on the boundary nodes. The material point velocity $v_{k,p}$ and coordinates $x_{k,p}$ are defined by mapping (i.e., an

95 interpolation) between the updated solution on the mesh and the material points, i.e.,

$$v_{k,p}^{t+\Delta t} = v_{k,p}^t + \Delta t \sum_{n=1}^{n_n} \phi_n(\boldsymbol{x}_p)a_{k,n}, \tag{9}$$

$$x_{k,p}^{t+\Delta t} = x_{k,p}^t + \Delta t \sum_{n=1}^{n_n} \phi_n(\boldsymbol{x}_p)v_{k,n}^{t+\Delta t}, \tag{10}$$

where $n_n$ is the number of associated nodes $n$ to a material point $p$. The remaining tasks are i) to update the material point volume and ii) to solve for the constitutive stress-strain relationship.

## 2.2 Rate formulation

The large deformation framework necessitates a suitable stress-strain formulation. Some studies prefer the finite deformation framework and employ a linear relationship between Kirchhoff stresses and logarithmic strains (Charlton et al., 2017; Gaume et al., 2018; Coombs et al., 2020). In the present work, we adopt a rate-dependent framework by applying the Jaumann rate (e.g., Huang et al. 2015; Wang et al. 2016c, b; Bandara et al. 2016), which yields an objective stress rate measure.

The Jaumann rate of the Cauchy stress is given by

$$\frac{\mathcal{D}\sigma_{ij}}{\mathcal{D}t} = C_{ijkl}\frac{1}{2}\left(\frac{\partial v_l}{\partial x_k} + \frac{\partial v_k}{\partial x_l}\right), \tag{11}$$

where $C_{ijkl}$ is the 4th rank tangent stiffness tensor. Thus, the Jaumann stress derivative may be written as

$$\frac{\mathcal{D}\sigma_{ij}}{\mathcal{D}t} = \frac{\mathrm{D}\sigma_{ij}}{\mathrm{D}t} - \sigma_{ik}\dot{\omega}_{jk} - \sigma_{jk}\dot{\omega}_{ik}, \tag{12}$$





where $\omega_{ij} = (\partial_i v_j - \partial_j v_i)/2$ is the vorticity tensor, $\mathrm{D}\sigma_{ij}/\mathrm{D}t$ corresponds to the material derivative

$$\frac{\mathrm{D}\sigma_{ij}}{\mathrm{D}t} = \frac{\partial \sigma_{ij}}{\partial t} + v_k \frac{\partial \sigma_{ij}}{\partial x_k}. \tag{13}$$

By rearranging the Jaumann stress derivative in Eq. 12, we obtain

$$\frac{\partial \sigma_{ij}}{\partial t} = \frac{\mathcal{D}\sigma_{ij}}{\mathcal{D}t} + \overbrace{\sigma_{ik}\dot{\omega}_{jk} + \sigma_{jk}\dot{\omega}_{ik}}^{\sigma_{ij}^{\mathcal{R}}}, \tag{14}$$

where $\sigma_{ij}^{\mathcal{R}}$ represents the rotation of the Cauchy stress tensor, which satisfies the stress objectivity for the rate-dependent formulation.

Let us expand $\sigma_{ij}^{\mathcal{R}}$ in Eq. 14 using identities $\sigma_{ij} = \sigma_{ji}$, $\dot{\omega}_{ij} = -\dot{\omega}_{ji}$ and $\dot{\omega}_{kk} = 0$. The Cauchy stress tensor is written using the so-called Voigt notation (as a vector $\boldsymbol{\sigma} = \{\sigma_{xx}, \sigma_{yy}, \sigma_{zz}, \sigma_{xy}, \sigma_{yz}, \sigma_{xz}\}$). After expanding, collecting and rearranging terms, the objective stress terms $\sigma_{ij}^{\mathcal{R}}$ for a three-dimensional configuration are

$$\sigma_{xx}^{\mathcal{R}} = 2(\sigma_{xy}\dot{\omega}_{xy} + \sigma_{xz}\dot{\omega}_{xz}), \tag{15}$$

$$\sigma_{yy}^{\mathcal{R}} = -2(\sigma_{xy}\dot{\omega}_{xy} - \sigma_{yz}\dot{\omega}_{yz}), \tag{16}$$

$$\sigma_{zz}^{\mathcal{R}} = -2(\sigma_{xz}\dot{\omega}_{xz} + \sigma_{yz}\dot{\omega}_{yz}), \tag{17}$$

$$\sigma_{xy}^{\mathcal{R}} = \dot{\omega}_{xy}(\sigma_{yy} - \sigma_{xx}) + \sigma_{yz}\dot{\omega}_{xz} + \sigma_{xz}\dot{\omega}_{yz}, \tag{18}$$

$$\sigma_{yz}^{\mathcal{R}} = \dot{\omega}_{yz}(\sigma_{zz} - \sigma_{yy}) - \sigma_{xy}\dot{\omega}_{xz} - \sigma_{xz}\dot{\omega}_{xy}, \tag{19}$$

$$\sigma_{xz}^{\mathcal{R}} = \dot{\omega}_{xz}(\sigma_{zz} - \sigma_{xx}) + \sigma_{yz}\dot{\omega}_{xy} - \sigma_{xy}\dot{\omega}_{yz}, \tag{20}$$

and, for a two-dimensional configuration assuming plane strain conditions, Eqs. 15, 16 and 18 reduce to

$$\sigma_{xx}^{\mathcal{R}} = 2\sigma_{xy}\dot{\omega}_{xy}, \tag{21}$$

$$\sigma_{yy}^{\mathcal{R}} = -2\sigma_{xy}\dot{\omega}_{xy}, \tag{22}$$

$$\sigma_{xy}^{\mathcal{R}} = \dot{\omega}_{xy}(\sigma_{yy} - \sigma_{xx}). \tag{23}$$

### 2.3 Elastoplastic deformation

A nonassociated Drucker-Prager model (D-P model) with a tension cutoff is used in this study, similar to Huang et al. (2015); Liu et al. (2020); Nguyen et al. (2020); Zuo et al. (2020), because of its straightforward implementation within explicit numerical solvers. The D-P model has been established as an approximation of the Mohr-Couloumb (M-C) model (Krabbenhoft et al., 2012; Alejano and Bobet, 2012), i.e., a conical yield surface that approximates the M-C yield surface in the principal stress space. The former can be adjusted by parameters, so it passes either through the outer or inner edges of the M-C yield surface (Jiang and Xie, 2011; De Borst et al., 2012).

The D-P yield function $f$ (see Fig. 1) is typically defined in terms of invariants; The first invariant of the Cauchy stress tensor $I_1 = \sigma_{kk}$, and the second invariant $J_2 = \frac{1}{2}\tau_{ij}\tau_{ji}$ of its deviatoric part $\tau_{ij}$, where the deviatoric part of the Cauchy stress





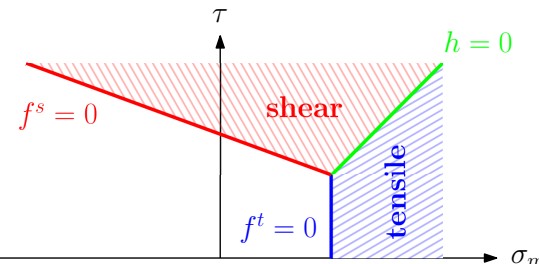

**Figure 1.** Drucker-Prager yield surface in the ($\sigma_m$-$\tau$) space. The yield surface is made of a shear line segment (in red) and a tensile line segment (in blue).

is $\tau_{ij} = \sigma_{ij} + \delta_{ij} p$ with the pressure $p = -\frac{1}{3}\sigma_{kk}$. The D-P yield surface is made of two surfaces (i.e., representing **s**hear and **t**ensile yield criteria), delimited by

$$f^s(\sigma_m, \tau) = \tau + q_\phi \sigma_m - k_\phi, \tag{24}$$

$$f^t(\sigma_m) = \sigma_m - \sigma^t, \tag{25}$$

where $\tau = \sqrt{J_2}$ is the effective shear stress, $\sigma_m = -p$ is the mean stress, $q_\phi$ and $k_\phi$ are the material parameters defined by $\phi$ as the internal friction angle, $\sigma^t$ is the tensile strength and $c$ is the cohesion. Cohesion varies with the accumulated plastic strain $\bar{\epsilon}_p$ when considering a strain softening material, i.e., $c = f(\bar{\epsilon}_p)$. These two surfaces define two plastic regions (see Fig. 1) corresponding to either the shear or tensile failure mode. We use a nonassociated plastic flow law for shear and tensile failures; thus, the plastic potential function $g$ is written as

$$g^s(\sigma_m, \tau) = \tau + q_\psi \sigma_m, \tag{26}$$

$$g^t(\sigma_m) = \sigma_m, \tag{27}$$

where $q_\psi$ is a material parameter estimated with the dilation angle $\psi$.

The line segment $h(\sigma_m, \tau) = 0$ represents the diagonal line between $f^s(\sigma_m, \tau) = 0$ and $f^t(\sigma_m, \tau) = 0$ in the ($\sigma_m, \tau$) plane, i.e., $h$ is the boundary between shear and tensile failure modes. The function $h(\sigma_m, \tau)$ is given by

$$h(\sigma_m, \tau) = \tau - \tau^P - \alpha^P(\sigma_m - \sigma^t), \tag{28}$$

with the constants $\tau^P = k_\phi - q_\phi \sigma^t$ and $\alpha^P = (1 - q_\phi^2)^{1/2} - q_\phi^2$. We consider an inner adjustment of the D-P yield surface with respect to the M-C yield surface (de Souza Neto et al., 2011), and the model parameter used in Eqs. 24 & 26 are given by

$$q_\phi = \frac{6\sin\phi}{\sqrt{3}(3 + \sin\phi)}, \tag{29}$$

$$q_\psi = \frac{6\sin\psi}{\sqrt{3}(3 + \sin\psi)}, \tag{30}$$

$$k_\phi = \frac{6c\cos\phi}{\sqrt{3}(3 + \sin\phi)}. \tag{31}$$





In the following, we briefly detail the return mapping strategy used to return the trial Cauchy stress $\sigma_{ij}^{tr}$ (i.e., assuming pure elastic deformation only) onto the yield surfaces considering $\psi = 0$. A complete description of such return mapping can be found in Huang et al. (2015). Shear failure is declared when i) $f^s(\sigma_m^{tr}, \tau^{tr}) > 0$ and $\sigma_m^{tr} < \sigma^t$ or if ii) $h(\sigma^{tr}, \tau^{tr}) > 0$ and

160 $\sigma_m^{tr} \geq \sigma^t$. The corrected Cauchy stress tensor now reads

$$\sigma_{ij}^{t+\Delta t} = \tau_{ij}^{tr} \left( \frac{k_\phi - q_\phi \sigma^{tr}}{\tau^{tr}} \right) + \sigma^{tr} \delta_{ij}, \tag{32}$$

with $\boldsymbol{\delta}$ the Kronecker tensor. Tensile failure is declared when $h(\sigma^{tr}, \tau^{tr}) \leq 0$ and $\sigma_m^{tr} \geq \sigma^t$. The corrected Cauchy stress tensor reads as

$$\sigma_{ij}^{t+\Delta t} = \sigma_{ij}^{tr} + (\sigma^t - \sigma_m^{tr}) \delta_{ij}. \tag{33}$$

## 165   3   GIMPM implementation under a GPU architecture

We propose an explicit generalized interpolation material point method (GIMPM) implementation (Dong and Grabe, 2018; Wang et al., 2020) in a three-dimensional configuration on a GPU, taking advantage of the efficient vectorized algorithmic structure (Wyser et al., 2020a, b). We select explicit GIMPM implementation, which is valid for a variety of problems compared to other latest variants (Wang et al., 2019; Coombs et al., 2020), i.e., CPDI or CPDI2q. Additionally, we use a double-mapping

approach (MUSL, see Nairn 2003; Buzzi et al. 2008), which consists of updating the stress at the end of the time step. We implement the following domain-update methods: a) no update of the material point domain, further labelled uGIMPM, and b) a domain update controlled by the determinant of the deformation gradient, i.e., $\det(F_{ij})$, further labelled cpGIMPM. These domain-update methods are commonly used in the literature (Baumgarten and Kamrin, 2019; Tran and Sołowski, 2019). The limitation of the two methods is that they are not ideally suited for specific tests: simple stretching and compression modes

(Coombs et al., 2020).

### 3.1   Implementation on a graphical processing unit (GPU)

Graphical processing units (GPUs) are many-core processors originally designed to refresh screen pixels (e.g., for computer games) independently. A schematic representation of the main architecture differences between a CPU and a GPU is depicted in Fig. 2. On the GPU chip, most of the physical space is dedicated to arithmetic logical units, whereas on a CPU, most of

180 the physical space is dedicated to chip host scheduling and control microsystems. GPUs feature many more cores, a lower thread-scheduling cost and a higher memory bandwidth than CPUs. The programming model is based on a parallel principle called single instruction - multiple data (or SIMD), i.e., every single instruction is executed on different data. GPUs feature a hierarchical structure. The lowest computational unit is the thread. Threads are organized into blocks of threads, the whole constituting a hierarchical grid of blocks of threads. A GPU typically launches thousands of threads, which execute the same

instruction in parallel, thus achieving massive parallelism. Additionally, the most recent GPUs offer a high throughput (close to a TB per second peak memory throughput).





Currently, most of the algorithms are memory-bounded, meaning that memory transfers limit the performance, in contrast to computer-bounded algorithms, where floating point (arithmetic) operations limit the performance. Thus, for an efficient implementation of an algorithm, one must consider a) limiting the memory transfers to the bare minimum and b) avoiding complex data structures (Räss et al., 2019a) to benefit from the high throughput capabilities of GPUs. The ability of a GPU is particularly well suited to efficiently execute a large number of local operations in parallel, i.e., single instruction, multiple data (SIMD) programming. In the case of a GIMPM implementation, this includes the calculation of shape functions and the update of various quantities at the material point level (i.e., stresses, domain lengths, material point volumes, etc.). Below, we present key aspects of our GPU-based implementation using the Computed Unified Device Architecture (CUDA C) language of the Nvidia Corporation, which is a syntax extension of the C programming language.

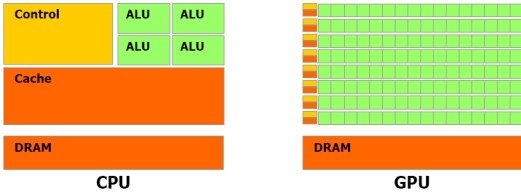

**Figure 2.** Schematic chip representation for both the central processing unit (CPU) and the graphical processing unit (GPU) architecture (Nvidia, 2007). The latter is made of thousands of arithmetic logical units (ALUs). The CPU architecture is primarily dedicated to controlling units and cache memory, and the physical space allowed for ALUs is considerably reduced compared to a GPU architecture.

### 3.1.1 Algorithm workflow

In our implementation, MATLAB acts as an architect (see Fig. 3). It 1) defines the problem geometry (i.e., the background mesh, material point locations and related quantities, etc.), which can be tedious to initialize in a CUDA C environment. It also calls an external MATLAB script, which compiles the necessary source codes, i.e., `gpu.cu` or `cpu.cu`. It further 2) calls either a CUDA C or plain C executable, i.e., `gpu.exe` or `cpu.exe`, within a Windows OS to solve for the numerical problem and finally 3) imports the results of calculations for further postprocessing tasks.

This is a powerful combination between a high-level language such as MATLAB and a performant low-level language such as CUDA C or plain C. It is also easy to invoke system commands directly via MATLAB, i.e., to compile source codes and/or run executables using the built-in command `system('...')`. We focus on OS-free scripting in MATLAB using a built-in command (i.e., `isunix` or `ispc`) to ensure that it performs well under all operating system (OS) architectures. In addition, such a workflow can be easily extended to other high-level languages such as Python.

### 3.1.2 Kernels and launch configuration

We briefly describe our GPU-based implementation (`gpu_main.cu`) while focusing mainly on the computational aspects of the implementation. Implementation of an explicit GIMPM solver into the CUDA C language requires dispatching computa-

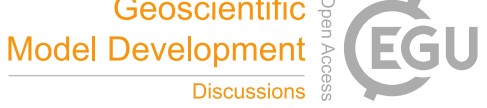



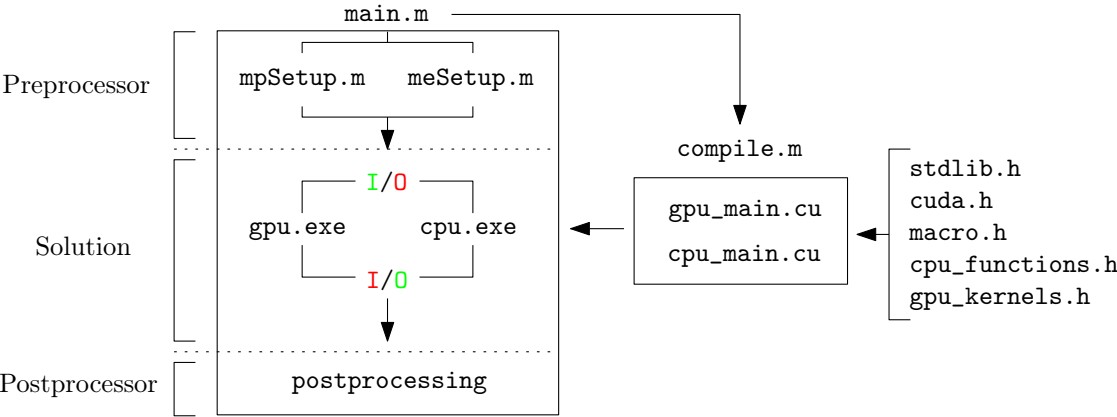

**Figure 3.** Multifunctional workflow: 1) usage of MATLAB for data initialization, compilation and postprocessing activities and 2) system calls to a performant compiled language such as C (CPU-based) and CUDA C (GPU-based) for heavy calculations. Here, `I/O` stands for input/output, and the colouring (red or green) specifies which one is active, i.e., `I/O` means data only are transferred to the GPU (or CPU) for further calculation activities.

tional activities into several kernels, i.e., similar to classic functions for a serial implementation in the C language. Each kernel is operated by the GPU only, and kernel launch configuration parameters must be defined for its proper execution. Among them, one must define the number of active threads per block (i.e., the block size) and the number of blocks (i.e., the grid size). A typical kernel is executed $N$ times in parallel by $N$ distinct threads organized into blocks of threads, i.e., a grid of blocks of threads. The principal hardware limitation is the total number of threads within a block: it cannot exceed 1024 threads per

block. One must ensure that the maximal size of a block is lower than or equal to this limit.

The computational activities are handled by multiple GPU kernels; 11 kernels are successively launched over a computational cycle. An overall description is given in Fig. 4. A `while` loop is used to perform the computational cycles, and an MPM step is solved at every cycle. $n_{\mathrm{IO}}$ (i.e., the number of accesses to the GPU global memory) is reported in Fig. 4 for each kernel and is estimated by a careful examination of relevant operations within the kernels. Note that all calculations are performed on the

GPU, except the calculation of the adaptive time step, which is serially executed by the CPU.

In our GPU-based implementation, we define two distinct types of kernel launch parameters: 1) those used for mapping between material points and background nodes (i.e., accumulations and projections between material points with their associated nodes and back and forth) and 2) those used for local calculation at the material point or node level (i.e., update of material point stresses or the solution to the momentum balance equations on the Eulerian background mesh).





```
                        ┌──── while t ≤ t_end ────┐
                        │  Function :   getdt(*)      │ ┐ cpu_functions.h
           g            │  Kernel 1 :   initD(*)      │   n_IO = 8n_no + 2n_el  ┐
           p            │  Kernel 2 :   basisD(*)     │   n_IO = n_mp(7n_n + 6)  │
           u            │  Kernel 3 :   accumD(*)     │   n_IO = n_mp(11 + 12n_n)│
           _            │  Kernel 4 :   solveD(*)     │   n_IO = 18n_no          │
           m            │  Kernel 5 :   projectD(*)   │   n_IO = 3n_mp(3 + n_n)  │  g
           a            │  Kernel 6 :   DMD(*)        │   n_IO = n_mp(4n_n + 3)  │  p
           i            │  Kernel 7 :   getdFD(*)     │   n_IO = 3n_mp(3 + 6n_n) │  u
           n            │  Kernel 8 :   elastD(*)     │   n_IO = 21n_mp          │  _
           .            │  Kernel 9 :   plastD(*)     │   n_IO ≠ cst             │  k
           c            │  Kernel 10:   volLockD1(*)  │   n_IO = 14n_mp          │  e
           u            │  Kernel 11:   volLockD2(*)  │   n_IO = 13n_mp          ┘  r
                        └─────────────────────────────┘                            n
                                                                                   e
                                                                                   l
                                                                                   s
                                                                                   .
                                                                                   h
```

**Figure 4.** Specific workflow for the source code running of the GPU, $t_{end}$ is a user-defined time that controls the total time of the simulation, and the operator $*$ stands for the pointer object, as in the C language. It should be noted that a vast majority of operations within kernels are performed on pointers.

### 3.1.3 Adaptative time step

An adaptive time step is implemented. For three-dimensional configurations, the maximum elastic wave speed of the material (Anderson, 1987; Zhang et al., 2017) reads as

$$(c_x, c_y, c_z) = c_{el} + \left( \max_p(|v_{x,p}|), \max_p(|v_{y,p}|), \max_p(|v_{z,p}|) \right), \tag{34}$$

where $c_{el} = ((K + 4G/3)/\rho)^{\frac{1}{2}}$ is the elastic wave speed of the material, $K$ and $G$ are the bulk and shear moduli, respectively, $\rho$ is the material density, and $v_{x,p}$, $v_{y,p}$ and $v_{z,p}$ are the material point velocity components. The time step $\Delta t$ is then restricted by the CFL condition,

$$\Delta t = \alpha \min \left( \frac{\Delta x}{c_x}, \frac{\Delta y}{c_y}, \frac{\Delta z}{c_z} \right), \tag{35}$$

where $\alpha \in [0; 1]$ is the time step multiplier, and $\Delta x, \Delta y$, and $\Delta z$ are the background mesh resolutions.

This requires evaluation of the maximum velocity of all material points at the beginning of each calculation cycle. We choose to sequentially find the maximum velocity using the CPU instead of a parallel implementation on the GPU. This results in systematic memory transfers between the GPU global memory and the random access memory (RAM) of the CPU. However, we report a low performance loss due to these transfers, i.e., a maximal loss of 2-5 % in performance, which is acceptable.

### 3.1.4 Back-and-forth mapping between material points and their associated nodes

The GPU-based algorithm relies heavily on the use of arrays p2e and e2n (Wyser et al., 2020a). Elements are numbered with an increasing index. Associated nodes are also numbered in a similar manner. The array e2n of dimension $n_{el} \times n_n$, where $n_{el}$ is the total number of nodes and $n_n$ is the number of nodes associated with an element $e$, describes the topological





relation between the elements and the nodes of the mesh. Similarly, the array `p2e` describes the topological relation between the material points and the element in which they are located. These two arrays provide an intuitive definition of the relations between i) the material points and the nodes they are associated with (i.e., `p2n`) and ii) the element and their nodes (i.e., `e2n`). Then, it is a computationally straightforward process to identify which nodes $n$ are associated with a material point $p$, which is occupying an element $e$.

The GPU-based implementation relies on the built-in function `atomicAdd()` in CUDA C. It performs atomic operations, which avoid the data race of multiple threads, from the same or different blocks to update the same memory location. Atomic operations are extensively used to calculate internal and external force contributions (Eqs. 6 & 7), as well as the lumped mass matrix, and to update the material point's properties such as velocities and coordinates (Eqs. 9 & 10). Dong et al. (2015a); Wang et al. (2020) reported (for older GPU architectures such as Pascal or Kepler) that atomic scattering can be significantly slower compared to an optimized parallel implementation. However, atomic operations are a) intuitive to both understand and implement, and b) they avoid a complex data layout, such as recently proposed in Wang et al. (2020). The use of built-in atomic operations considerably reduces programming efforts.

### 3.1.5 Treatment of volumetric locking for low-order elements

When low-order elements are used in a GIMP formulation, volumetric locking arises and results in spurious oscillations of the stress field (Jassim et al., 2013; Coombs et al., 2018; González Acosta et al., 2019; González Acosta et al., 2021). We implement a simple procedure to mitigate volumetric locking when considering near-incompressible behaviour for ischoric plastic flows. Cuomo et al. (2019); Lei et al. (2020) introduced an element-based averaging method, following Mast et al. (2012). Selected material point properties are reconstructed based on an average value calculated at the element's centre at the end of a time step. However, we propose averaging only the volumetric part of the stress tensor, i.e., the pressure $p = -\frac{1}{3}\sigma_{kk}$, while its deviatoric part $\tau_{ij} = \sigma_{ij} - p\delta_{ij}$ remains unchanged. This results in the following:

$$p_e = \frac{\sum_{p \in e} v_p p_p}{\sum_{p \in e} v_p}, \tag{36}$$

where $v_p$ is the material point's volume. This gives a constant distribution of the pressure field over an element because of its zero-order reconstruction (Lei et al., 2020). The Cauchy stress tensor $\sigma_{ij,p}$ of a material point $p$ occupying an element $e$ is corrected as

$$\sigma_{ij,p} = \tau_{ij,p} + \delta_{ij}(p_e)_p, \tag{37}$$

where $\delta_{ij}$ is the Kronecker delta and $(p_e)_p$ is the averaged pressure within an element $e$ and assigned to a material point $p$.

### 3.2 Available computational resources

The CPU- and GPU-based simulations are performed on a modern workstation running on a Windows 10 operating system with the latest CUDA version v11.2. The CPU is an Intel Core i9-10900K with 10 physical cores of base clock speed (or





frequency) of 3.70 GHz, which can rise up to a maximum clock speed of 5.30 GHz, supported with 64 GB DDR4 RAM. It hosts a consumer electronics Nvidia RTX 3090 GPU (the latest Ampere architecture) with 82 streaming multiprocessors (SM units) with a base frequency of 1.40 GHz. This results in 10490 CUDA cores that are supported with an on-chip memory of 24

GB GDDR6 (i.e., the GPU global memory). Other GPUs installed on older desktops are also used to compare their respective GPU performances, i.e., an RTX 2080 ti (workstation) and a GTX 1650 (laptop), both running on a Windows 10 operating system. Additional simulations were also ran on a workstation equipped with the latest Nvidia A100 GPU at the Lomonosov Moscow State University.

Furthermore, GPU-based simulations are also performed on the Octopus GPU supercomputer at the Swiss Geocomputing

Centre, University of Lausanne, Switzerland. In particular, the GPU-based simulations are run on the Volta node, hosting an Nvidia Tesla V100 (Volta architecture) 16 GB, supported by an Intel(R) Xeon(R) E5-2620 v2 (Haswell) @ 2.1 GHz CPU. The latest CUDA version installed is v11.0, and the supercomputer Octopus is operated under a CentOS 6.9. environment. To summarize the computational resources in use, Table 1 presents the main characteristics of the GPUs used in this study.

**Table 1.** List of the graphical processing units (GPUs) used throughout this study. We also report the peak memory throughput, i.e., $\text{MTP}_{\text{peak}}$, measured thanks to the routine `bandwidthTest.cu` provided by Nvidia alongside with the CUDA toolkit. When compared with the effective memory throughput $\text{MTP}_{\text{eff}}$, one can estimate the possible gain of an additional optimization of the algorithm. This is particularly useful when estimating the level of optimization of a GPU-based implementation.

| GPU | Architecture | SM count | On-chip memory [GB] | $\text{MTP}_{\text{peak}}$ [GB·s$^{-1}$] |
|---|---|---|---|---|
| A100 | Ampere | 108 | 40 | 1127.1 |
| RTX 3090 | Ampere | 82 | 24 | 774.1 |
| RTX 2080 ti | Turing | 68 | 11 | 513.1 |
| GTX 1650 | Turing | 14 | 4 | 168.7 |
| V100 | Volta | 80 | 16 | 732.6 |

### 3.3 Measuring computational performance on a GPU

Omlin (2017); Räss et al. (2019a, b); Alkhimenkov et al. (2021) demonstrated that a pertinent metric to quantify the performance of memory-bounded algorithms is the effective memory throughput, i.e., $\text{MTP}_{\text{eff}}$ in GB·s$^{-1}$. It quantifies the efficiency of data transfers between the global memory (i.e., the on-chip memory of the GPU) and the arithmetic logical units (ALUs) of the GPU. To determine the effective memory throughput, one must estimates (or quantifies) the overall set of memory operations (read-and-write or read only), i.e., $n_{\text{IO}}$, which are needed to resolve a given problem. Consequently, we carefully estimate

the minimum number of memory operations while considering a GIMPM-based implementation. This results in the following effective memory throughput:

$$\text{MTP}_{\text{eff}} = \frac{n_{\text{iter}} \times n_{\text{IO}} \times n_{\text{p}}}{1024^3 \times t_{\text{GPU}}} \ [\text{GB} \cdot \text{s}^{-1}], \tag{38}$$





where $n_{\mathrm{p}}$ is the arithmetic precision (i.e., single-precision floating-point format FP32 or double-precision floating-point format FP64) and $t_{\mathrm{GPU}}$ is the wall-clock time in seconds to complete the $n_{\mathrm{iter}}$ iterations to solve for the numerical problem. For three-dimensional problems, we estimate the minimal number of memory operations for an explicit GIMP implementation as

$$n_{\mathrm{IO}} = 2n_{mp}(43 + 22n_n) + 26n_{no} + 2n_{el}, \tag{39}$$

where $n_{mp}$ is the number of material points, $n_n$ is the number of associated nodes for an element (i.e., $n_n = 16$ in 2D and $n_n = 64$ in 3D), $n_{no}$ is the number of nodes, and $n_{el}$ is the number of elements. Additionally, we also report the count of calculation cycles per second of the GPU, i.e., it·s$^{-1}$ as well as the wall-clock time. These two metrics give an intuitive sense of the time-to-solution, which is convenient for potential application purposes.

## 4 Results

In this section, we present two numerical models using the solver ep2-3De v1.0, namely,

1. Model 1, the granular collapse, which serves as

    (a) a validation benchmark against the results of the widely-accepted experiment of Bui et al. (2008) under a three-dimensional configuration

    (b) a demonstration of the influence of the mesh resolution on plastic strain localization under a plane strain configuration

2. Model 2, the three-dimensional earth slump (Varnes, 1958, 1978), which serves as

    (a) an evaluation of the relative performances of GPU- and CPU-based implementation of the solver ep2-3De v1.0 considering a variety of recent GPU architectures

    (b) a showcase of a potential application of the solver ep2-3De v1.0 for an elastoplastic problem considering different isotropic peak cohesion fields (homogeneous and heterogeneous)

### 4.1 Model 1

#### 4.1.1 Settings for Models 1a & 1b

We investigate the granular collapse of an aluminium-bar assemblage (Bui et al., 2008) under three-dimensional or plane strain configurations. The geometry of the problem is shown in Fig. 5, and its variables are summarized in Table 2 for both three-dimensional and plane strain configurations. Note that for Model 1a, we use the same number of elements along the $x-$direction $n_{el,x} = 80$ as in Huang et al. (2015). As a direct comparison for Model 1b under a plane strain configuration, Huang et al. (2015) used $n_{el} = 15360$, $\Delta x = \Delta z = 2.5\,\mathrm{mm}$ and $n_{mp} = 25600$.

We consider a noncohesive granular material of density $\rho = 2650\,\mathrm{kg \cdot m^{-3}}$, with a bulk modulus $K = 0.7\,\mathrm{MPa}$ and a Poisson's ratio $\nu = 0.3$, as in Huang et al. (2015). The cohesion is $c = 0\,\mathrm{Pa}$, the internal friction angle is $\phi = 19.8°$ with a dilatancy angle





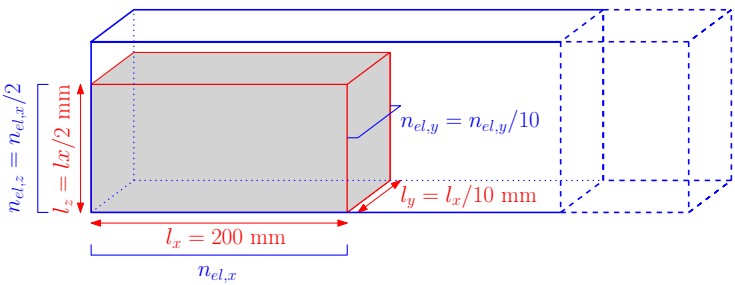

**Figure 5.** Initial configuration for the granular collapse numerical model. The blue surrounding frame depicts the computational domain, i.e., the background Eulerian mesh, and the red volume is the granular material, which is discretized by 8 material points. The total number of background elements $n_{el}$ depends on the number of elements in the $x-$ direction $n_{el,x}$ used to discretize the granular material.

$\psi = 0$ according to Bui et al. (2008). However, the density and stiffness properties have negligible effects on the granular flow dynamics, as reported by Nguyen et al. (2020). We introduce local damping $D$ (see Wang et al. 2016b) to resolve numerical results that are compatible with the experimental results of Bui et al. (2008). We find that $D = 0.025$ results in the most

compatible dynamics. The reasons for the introduction of local damping can be found in Appendix C. Fully fixed boundary conditions (i.e., no slip) are enforced at the bottom and rollers on the sidewalls. The total simulation time is 1.0 s, considering a the time step multiplier $\alpha = 0.5$.

**Table 2.** Parameters used in Models 1 a & b for the granular collapse. $n_{el,i}$ is the number of elements to discretize the granular material along the i-th direction, $n_{el}$ and $n_{no}$ are the total number of elements and nodes of the background mesh, $n_{pe}$ is the number of material points per element and $n_{mp}$ is the total number of material points. Note that the mesh resolution is $\Delta x = \Delta y = \Delta z = 2.5$ mm.

| Experiment | $n_{el,x}$ | $n_{el,y}$ | $n_{el,z}$ | $n_{el}$ | $n_{no}$ | $n_{pe}$ | $n_{mp}$ | $\Delta x$ [mm] |
|---|---|---|---|---|---|---|---|---|
| 1a | 80 | 20 | 40 | 342144 | 365625 | 8 | 512000 | 2.5 |
| 1b | 640 | - | 240 | 833300 | 836190 | 4 | 819200 | 0.3 |

### 4.1.2  Model 1a: the three-dimensional granular collapse

To validate the numerical implementation under a GPU architecture, we first compare it against the well-known granular

collapse experiments initially performed by Bui et al. (2008). Here, we present and compare numerical results without focusing on the performance of the GPU-based implementation. All the simulations are performed on a consumer electronics RTX 3090 GPU with double-arithmetic precision (i.e., $n_p = 8$ bytes).

The results from the numerical simulation under a three-dimensional configuration are shown in Fig. 6. A direct and visual comparison demonstrates excellent agreement between the numerical solver and the experiments of Bui et al. (2008). We

observe a slightly higher run-out distance, but the overall geometry of both the failure surface and the free surface is very close to the experimental data. We also report an angle of repose of $\approx 13°$. This value is also consistent with the value reported by



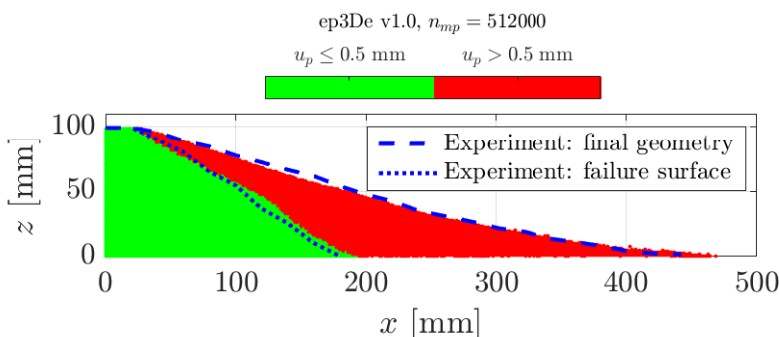

**Figure 6.** Final geometry of the granular collapse for three-dimensional configuration of our GPU-based explicit GIMPM implementation ep3De v1.0. The green region (i.e., the intact region) is defined by the $L_2$-norm of the material point displacement $u_p = ||\boldsymbol{u}_p||_2 \leq 0.5$ mm, whereas the red region (i.e., the deformed region) is defined by $u_p = ||\boldsymbol{u}_p||_2 > 0.5$ mm. The experiment of Bui et al. (2008) is indicated by the blue dashed line (i.e., the free surface) and the blue dotted line (i.e., the failure surface).

Bui et al. (2008), i.e., $14°$. The good agreement between the numerical results and the experimental work of Bui et al. (2008) demonstrates that the solver ep2-3De v1.0 is suitable to simulate large deformation elastoplastic problems such as granular collapses.

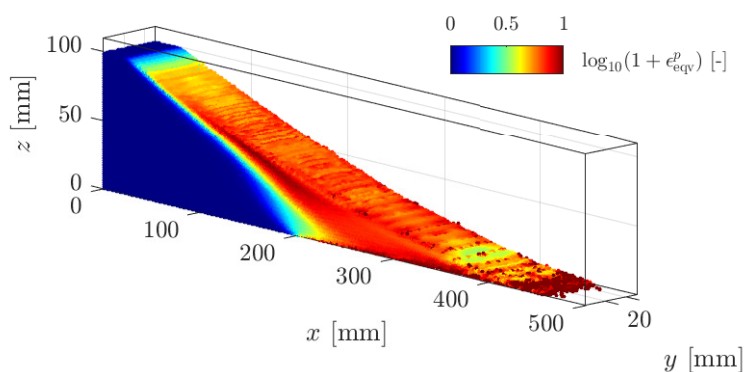

**Figure 7.** Equivalent plastic strain $\epsilon_{\text{eqv}}^p$ for the final configuration of the granular collapse. The principal feature of a granular collapse can be observed, i.e., a backward propagation of plastic deformation along a principal failure surface.

The equivalent accumulated plastic strain $\epsilon_{\text{eqv}}^p$ is shown in Fig. 7. We observe a coherent deformation of the granular material with a large shear zone that propagates backward from the base of the material to the top of the granular material. The mobilized granular material flows along a principal failure surface. However, the overall deformation pattern is rather coarse, i.e., fine structures or local shear bands are not yet observed, even though slight deformation heterogeneities can be observed. This coarse behaviour of shear banding is also consistent with previous studies (see Huang et al. 2015; Chalk et al. 2020; Zhang





et al. 2021). This is mainly due to the background mesh resolution used in the numerical simulation. We further investigate shear banding using a higher background mesh resolution under a plane strain configuration in Model 1b.

### 4.1.3 Model 1b: the plane strain granular collapse

We investigate granular collapse under a plane strain configuration, as this allows an increase in the number of elements, resulting in an even finer background mesh (see Table 2). For Model 1a, the numerical solution is in agreement with the
350 experimental work of Bui et al. (2008) regarding either the free surface or the failure surface (see Fig. 8). This demonstrates that both the three-dimensional and plane strain configurations are in agreement with each other.

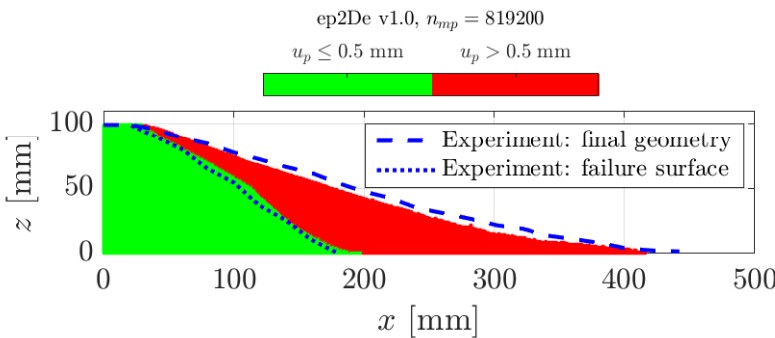

**Figure 8.** Final geometry of the granular collapse for the plane-strain configuration for our GPU-based explicit GIMPM implementation ep2De v1.0. The numerical solution and the experimental results are in good agreement. Some differences are more pronounced when compared with the numerical results obtained under a three-dimensional configuration.

An interesting feature of granular collapse is the equivalent accumulated plastic strain (see Fig. 9). The GPU-based implementation allows us to increase both the background mesh resolution and the total number of material points. This results in finer plastic strain localizations, as demonstrated by the various shear bands and their complex interactions during collapse.
Such detailed shear bands are almost impossible to obtain at lower resolutions, which demonstrates the importance of a GPU-based implementation to overcome the hardware limitation of a CPU-based implementation, i.e., mainly longer wall-clock times.

### 4.2 Model 2

#### 4.2.1 Settings for Models 2a & 2b

Here, we select a cohesive elastoplastic isotropic material (i.e., a homogeneous or heterogeneous peak cohesion field) with no dilatancy behaviour. It is modelled with a pressure-sensitive Drucker-Prager model with linear strain-softening behaviour. It is well known that the numerical solutions (as in FEM) are mesh-dependent when considering the strain-softening behaviour of



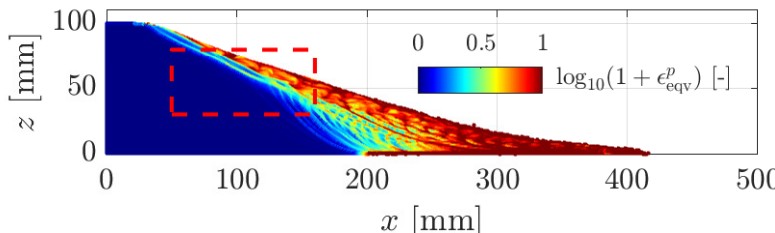

**Figure 9.** Equivalent plastic strain $\epsilon^p_{\mathrm{eqv}}$ for the final configuration of the granular collapse. The dashed red rectangle denotes the location of the zoomed-in region in Fig. 10. One can observe more complex plastic strain localizations compared to the numerical results obtained in Fig. 7 for a three-dimensional configuration with a coarser background mesh resolution.

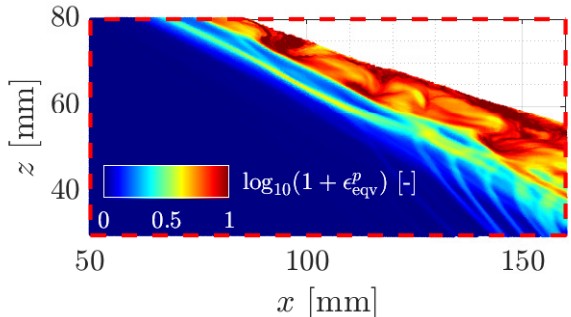

**Figure 10.** Equivalent plastic strain $\epsilon^p_{\mathrm{eqv}}$ for the zoomed-in area in Fig. 9. A shallow granular flow clearly appears, as suggested by the higher values of $\epsilon^p_{\mathrm{eqv}}$. This supports evidence of shallower granular avalanches during collapses. Deeper structures, which result in lower accumulated plastic strains, probably highlight slower deformation modes along well-defined and persistent shear bands.

the material. We did not implement techniques to address this issue, but the use of nonlocal plasticity (Galavi and Schweiger, 2010; Burghardt et al., 2012) or viscoplastic formulations (Duretz et al., 2019) are possible ways to address this specific task.

We have chosen an arbitrary geometry (see Fig. 11 and Table 4), which represents an idealized three-dimensional setting, to observe elastoplastic *slumps* (i.e., earth slumps according to the original classification proposed by Varnes 1958, 1978), which are now classified as rotational slides in the recent update of the Varnes classification proposed by Hungr et al. (2014). The geometrical setting differs from the one typically used in the literature, as in Zhang et al. (2021). However, it promotes the compression of the toe, which is an expected feature we want to reproduce. The size of the physical domain $l_z \times l_x \times l_y$ is, at

most, 12 m × 64 m × 1024 m for Model 2a, whereas it is 12 m × 64 m × 16 m for Model 2b.

We assume this setting features the principal first-order characteristics of a typical rotational earth slump (Varnes, 1958, 1978), i.e., a complex zone of scarps (minor and major) delimiting a crown-like structure, followed by a transition (or depletion) zone in which the material flows homogeneously along internal shear zones due to severe plastic strain localizations and, finally, a compression (or accumulation) zone resulting in complex thrusting at the toe of the slump. Because of the nature of





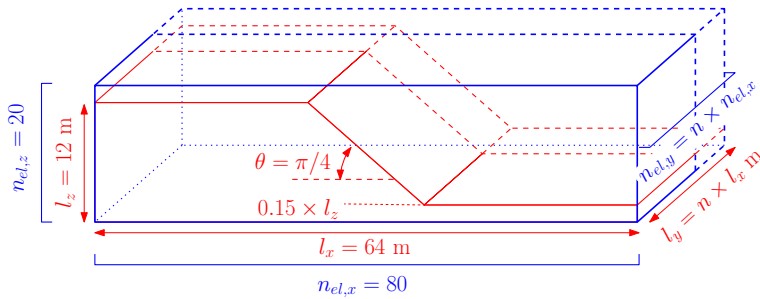

**Figure 11.** Geometry for the earth slump. The number of elements in the $y$-direction $n_{el,y}$ and the width of the problem $l_y$ are variable. This allows us to increase (or decrease) the number of both elements and material points without decreasing the mesh resolution. The parameter $n$ controls the dimension of the domain and the number of elements along the $y-$direction. The wall-clock time depends only on the total number of elements, nodes and material points and is not influenced by the mesh resolution.

the boundary condition at the bottom of the material (i.e., free-slip), an additional horizontal sliding component is introduced within the rotational part of the displacement.

**Table 3.** Material properties shared by both Models 2a & 2b.

| Parameter | Symbol | Value | Unit |
|---|---|---|---|
| Density | $\rho$ | 2700 | kg·m$^{-3}$ |
| Poisson's ratio | $\nu$ | 0.3 | - |
| Elastic modulus | $E$ | 1 | MPa |
| Softening modulus | $H$ | 50 | kPa |
| Friction angles | $\phi$ / $\phi_{\text{weak}}$ | 20 / 7.5 | ° |

We select material properties (i.e., bulk and shear moduli $K$ and $G$, friction angle $\phi$ and peak and residual cohesions $c_{\text{peak}}$ and $c_{\text{res}}$) that result in severe deformation processes and strain localizations. The material properties are presented in Table 3. They are close to the values commonly used in the literature (Wang et al., 2016b, a; Bandara et al., 2016; Zhang et al.,

2021). To increase deformations even more, we also introduce a weak layer of thickness $0.3 \times l_z$ m at the base of the material with a lower friction angle $\phi_{\text{weak}}$. A time step multiplier $\alpha = 0.5$ is selected, i.e., $\Delta t_{min} = 1.56 \cdot 10^{-2}$ s is obtained over the whole simulation according to the CFL condition for both Models 2a & 2b. As in Zhang et al. (2021), elastic loading dynamic relaxation is applied for a period of $t = 8$ s (i.e., Models 2a & 2b), and the elastoplastic behaviour is activated for an additional 7 s, resulting in a total simulation time $t = 15$ s (i.e., Model 2b only).

Gaussian random fields (see Appendix B) are used to initialize the peak cohesion field $c_{\text{peak}}$, which is parametrized by an average cohesion $\bar{c}_{peak}$ and its standard deviation $\sigma$ (see Table 4) along with the residual cohesion $c_{\text{res}} = c_{\text{peak}}/4$. This allows us to account for heterogeneities within the material, which lead to complex and heterogeneous displacement fields. We first



**Table 4.** Geometrical and material properties for Models 2a & 2b. The correlation length vector is $\boldsymbol{\lambda} = (\lambda_x, \lambda_y, \lambda_z) = (2.5, 2.5, 2.5)$ m for both Gaussian and exponential isotropic covariance functions. The grid spacing is always constant in Models 2a & 2b, i.e., $\Delta z = \Delta y = \Delta x = 0.8$ m

| Model | $n_{el,y}$ [-] | $n_{mp}$ [-] | $\Delta x$ [m] | $\bar{c}_{\text{peak}}$ [kPa] | $\sigma$ [kPa] |
|-------|------------|-----------|------------|--------------------|-----------|
| 2a | $\in [1; 1280]$ | $\leq 3.2 \cdot 10^6$ | 0.8 | 20 | 0 |
| 2b | 20 | $\approx 10^5$ | 0.8 | 20 | 0 / 5 |

perform preliminary simulations with a constant cohesion field and notice a homogenous solution of the displacement field along the $y-$direction. Using Gaussian fields allows us to mitigate this homogeneity.

Free-slip boundary conditions are applied on the sides and the bottom of the computational domain; only the normal component to the boundary is constrained, while the two others are free. This results in stronger deformations, which we want to highlight. Finally, and as suggested in Wang et al. (2016b), we introduce local damping, i.e., $D = 0.1$.

### 4.2.2    Model 2a: relative performances

Here, we investigate the computational performances of the solver ep2-3De v1.0 under a three-dimensional configuration on a
variety of GPUs with recent architectures: Ampere, Turing and Volta. Furthermore, we restrict our performance analysis only for the elastic loading phase (i.e., 8 s of simulation) because it is more complex to determine the exact number of material points that are yielding during each computational cycle (see Fig. 4) and to infer the exact effective memory throughput.

All the numerical simulations are performed on the computational resources and GPU hardware presented in Table 1 under double-arithmetic precision (i.e., $n_{\text{p}} = 8$ bytes in Eq. 38). As a reference baseline, we use the performance obtained for a
CPU-based single-threaded implementation of ep2-3De v1.0 on an i9-10900K CPU (e.g., latest Intel CPU chip). However, this is not representative of a highly optimized multithreaded implementation under a CPU architecture.

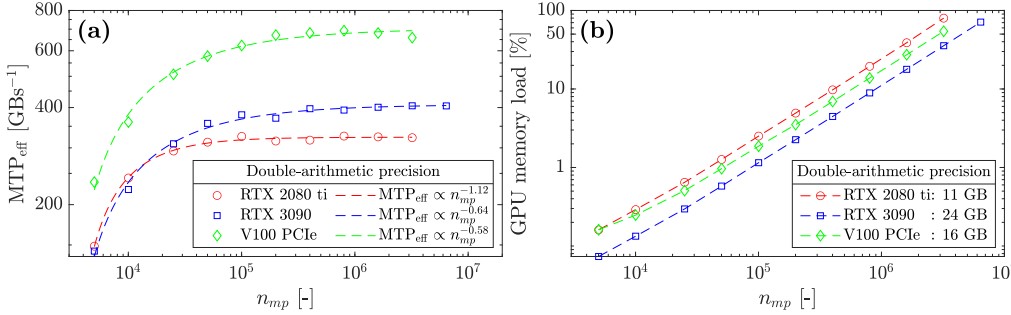

**Figure 12.** a) Effective memory throughput MTP$_{\text{eff}}$ of the solver ep2-3De v1.0 for double-arithmetic precision. One can see the on-chip memory limit, as both the RTX 2080 ti and V100 cannot resolve the same number of material points as the RTX 3090. b) GPU on-chip memory load increases with the number of material points $n_{mp}$, which demonstrates, as expected, one of the GPU's hardware limits.



We report the effective memory throughput $MTP_{eff}$ of the solver ep2-3De v1.0 on various GPUs and CPUs (see Fig. 12). An increase in the effective memory throughput is observed as the number of material points increases. All GPUs reach a maximum effective throughput, but the Tesla V100 scores a maximum effective throughput of $\approx 650\,\mathrm{GB\cdot s^{-1}}$. This corresponds to 88 %

of its peak throughput (for the GPU's hardware limit, see Table 1). We report a similar observation for the RTX 2080 ti, $MTP_{eff} \approx 320\,\mathrm{GB\cdot s^{-1}}$ corresponding to 62 % of its hardware limit. RTX 3090 and GTX 1650 reach $MTP_{eff} \approx 405\,\mathrm{GB\cdot s^{-1}}$ and $MTP_{eff} \approx 75\,\mathrm{GB\cdot s^{-1}}$, respectively, which correspond to 52 % and 44 % of their respective hardware limits. Finally, we report a memory throughput of at least $MTP_{eff} \approx 5\,\mathrm{GB\cdot s^{-1}}$ for the i9-10900K CPU (10 % of its hardware limit).

The overall results suggest, as in Räss et al. (2019b), that most recent GPUs, such as the data-centre Tesla V100 (Volta), offer

significant performances compared to entry-level consumer electronics GPUs, such as the GTX 1650. In terms of absolute performance, the more recent the GPU is, the higher its performance. A demonstration is given by the absolute effective throughput between the RTX 2080 ti and the RTX 3090: The latter achieves an additional 20 % throughput compared to the former. We highly suspect the hardware itself to be the main reason for this. We further investigate the performances of the most recent data-centre GPU, i.e., the A100 (Ampere architecture), with its predecessor the V100 (Tesla architecture). The

A100 reaches $\approx 1100\,\mathrm{GB\cdot s^{-1}}$, which yields a performance gain of $1.6\times$ with respect to the Tesla V100. When compared to the maximum effective memory throughput in Table 1, this correspond to 97 % of the hardware limit.

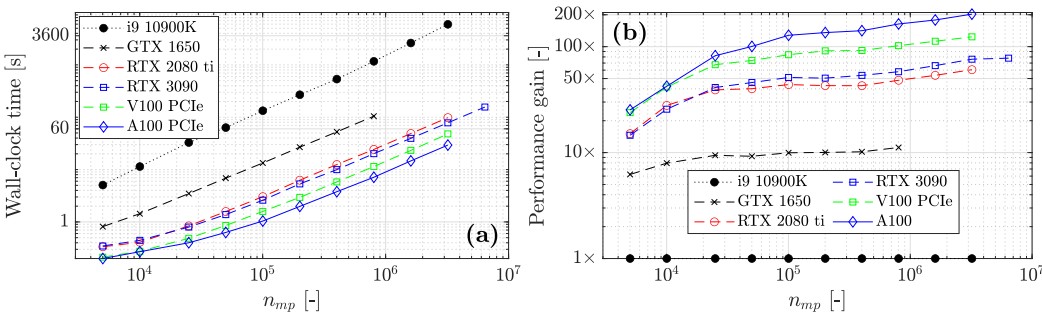

**Figure 13.** a) Wall-clock time reported for various computing architectures (GPUs and CPU). The differences in the maximal number of material points $n_{mp}$ are due to the on-chip memory limit. A significant difference in terms of wall-clock time is observed between the CPU and GPUs, even for the low-entry consumer electronic GTX 1650, i.e., a performance gain of $\approx 10\times$. b) Performance gains of GPUs relative to the CPU, i.e., $1\times$ as a baseline. We add the CPU and the GTX 1650 wall-clock time for an easier comparison.

Finally, we report the wall-clock time for various computing architectures (see Fig. 13a). As expected by the maximum effective memory throughput, A100 delivers the fastest solution, regardless of the number of material points $n_{mp}$. The A100 GPU resolves a geometry of $n_{mp} \approx 3.2 \cdot 10^6$ in less than a minute (29 seconds), whereas the i9-10900K CPU resolves the

same problem in more than an hour (5949 seconds). This corresponds to a $200\times$ performance gain ($123\times$ performance gain for the V100, see Fig. 13b) compared to the CPU-based implementation of ep2-3De v1.0. The RTX 2080 ti and the RTX 3090 reach a $60\times$ and $77\times$ performance gain, respectively. However, the entry-level GTX 1650 is only ten times faster than i9-10900 K. As already shown in Fig. 12a, these performance gains are only expected when the different GPUs reach their





maximum effective memory throughput. In terms of runtime, the performance gain (Fig. 13b) is in agreement with the memory
throughputs reported in Fig. 12a.

### 4.2.3    Model 2b: homogeneous and heterogeneous slumps

As a final experiment, we show the results of the ep2-3De v1.0 solver for a slump with homogeneous or heterogeneous cohesion
fields. In this numerical model, we only show the displacement field at the end of the numerical simulation at $t = 15$ s. The
interested reader is referred to Appendix D for an overview of the temporal evolution of the equivalent plastic strain $\epsilon_{\mathrm{eqv}}$ for
the slump under the three settings of the peak cohesion field. All the numerical simulations are run on a laptop equipped with
GTX 1650; $t_{\mathrm{GPU}} \approx 30$ s with the settings presented in Table 4. In the following, we present the main results for the three peak
cohesion fields, and we discuss the main characteristics obtained for typical slumping mechanics.

### 4.2.4    Homogeneous peak cohesion field

The homogeneous solution gives preliminarily interesting results (see Fig. 14). The first-order characteristics of a slump can
be observed, even though their magnitude is relatively fair compared to the real slump. The most striking feature is the devel-
opment of one major shear zone, along which the material flows (i.e., depletion) towards the toe of the slump, resulting in a
compression zone (i.e., thrusting and folding deformations). The crown-like structure develops linearly along the $y-$direction
and is highly localized at the surface of the slump (at $x \approx 20$ m in Fig. 14). However, the material flows homogeneously along
the $x-$direction (see the vertical profile in Fig. 14), as shown by the displacement field. The lateral variation of the displace-
ment field (along the $y-$direction) is almost nonexistent, which is mainly due to the spatial homogeneity of the peak cohesion
field.

### 4.2.5    Isotropic Gaussian covariance

Considering heterogeneities with a Gaussian covariance function for the cohesion field, the displacement field starts to resolve
a differential behaviour (see Fig. 15). Higher and/or weaker values of the peak cohesion field yield lower and/or greater dis-
placements. This is obvious, especially in the transition zone where this differential is observable. In addition, the compression
zone also starts to resolve spatial variations due to weaker and stronger cohesion values.

A striking difference is the shear zone itself (see Fig. D2): the shear zone exhibits a more complex spatial pattern, whereas
only one major shear zone is observed in Fig. D2. Retrogressive shear banding appears during the time evolution of the
slump, which suggests the development of a secondary shear zone within the slump. Moreover, the crown-like structure is now
curved and not linear along the $y-$direction. Its spatial extent is more important and is not as localized as in the homogeneous
case. Nevertheless, a more complex arrangement of major and minor scarps within the crown-like structure has not yet been
observed. Such a structure is more evident if one observes the accumulated equivalent plastic strain $\epsilon_{\mathrm{eqv}}^{p}$ in Fig. D2 in Appendix
D.



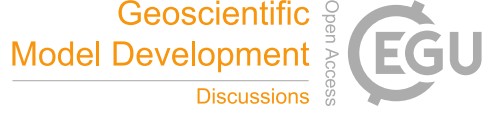

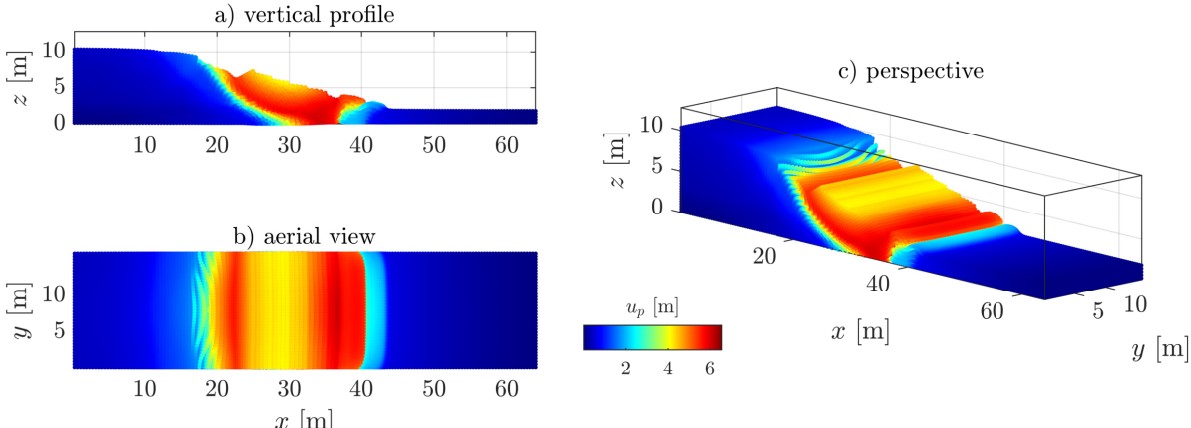

**Figure 14.** Displacement field obtained after $t = 15$ s for a homogeneous peak cohesion field. One can see an overall homogenous displacement field with some of the first-order characteristics of a slump, i.e., a rotational displacement with a compression zone at the toe, a transition zone delimited by one principal shear zone and a major scarp at the top of the material.

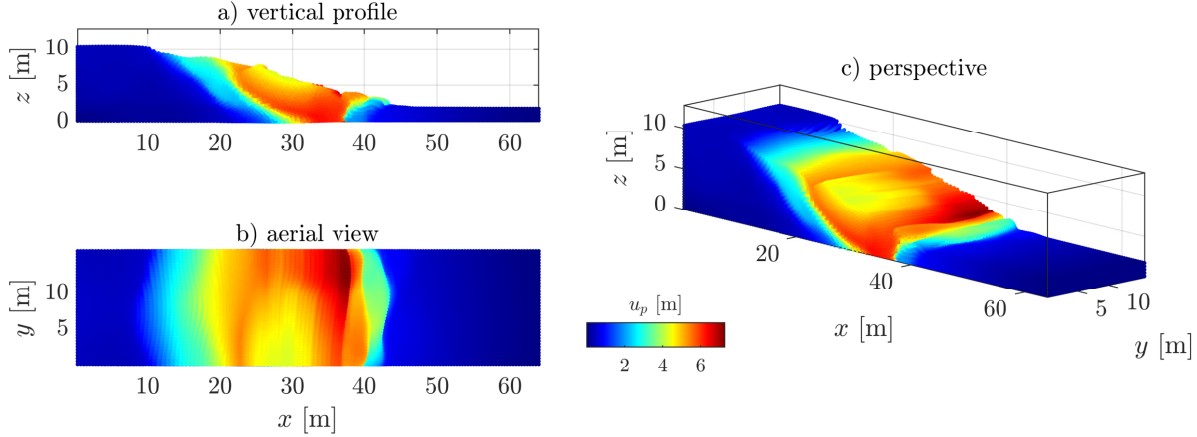

**Figure 15.** Displacement field obtained after $t = 15$ s for a heterogeneous peak cohesion field with a Gaussian covariance function.





The high magnitude of the displacement field in the areas $x \in [20; 40]$ m and $y \geq 8$ m is due to a weaker zone in the peak
cohesion field (see Fig. D2). This shows a strong influence of the heterogeneous peak cohesion field on the final displacement
field. A lower shear strength of the material yields faster strain-softening behaviour, promoting a faster response of shear
banding.

### 4.2.6 Isotropic exponential covariance

Shear banding activities become even more complex when an exponential covariance function is used, relative to Fig. 14 and
even with Fig. 15 to some extent. The spatial distribution of the peak cohesion (see Fig. D3) resolves finer heterogeneities with
a smaller length scale compared to when Gaussian covariance is used. Principal differences are observed at the top and toe
of the slump, where the crown-like structure turns into a complex zone made of minor and major scarps (see Fig. 16). The
displacement field becomes highly heterogeneous, particularly at the toe and the top of the slump. However, it is also more
homogeneous when compared with Fig. 15, particularly in $x \in [25; 35]$. The difference is evident between Figs. 17 & 15 at this
particular location.

The difference between the Gaussian and exponential covariance of the peak cohesion suggests the following. Heterogeneous
displacement fields could be influenced by larger and/or coarser fluctuations of the shear strength within the material. By
extrapolation, this could imply that the magnitude of the heterogeneity might be related to the fluctuation scales of the peak
cohesion field. Locally rather homogeneous fluctuations of the peak cohesion (i.e., Gaussian covariance) seem to promote
an increasingly heterogeneous displacement field at the surface. The characteristic length scale of spatial fluctuations could
have important implications for highly heterogeneous displacements within landslides. The same assumption could hold for
understanding the more complex crown-like structure of slumps (see Fig. D3)

## 5 Discussion

### 5.1 GIMPM suitability

We investigated granular collapses in both three-dimensional and plane strain configurations. Our numerical results demon-
strated the suitability of GIMPM to correctly reproduce experimental granular collapses. They also demonstrated that the
results did not significantly differ between these two spatial configurations and that both approaches give similar numerical
solutions.

### 5.2 Collapse limitation

For Model 1a, the principal hardware limit is the on-chip memory of the GPU. Even though RTX3090 is supported by 24
GB DDR4, it is physically impossible to achieve the resolution used for plane strain granular collapse. This would require
more than 24 GB of on-chip memory. However, a multi-GPU implementation using the message passing interface API (MPI)
could resolve this hardware limitation when using only one GPU. As demonstrated by Räss et al. (2019a); Alkhimenkov





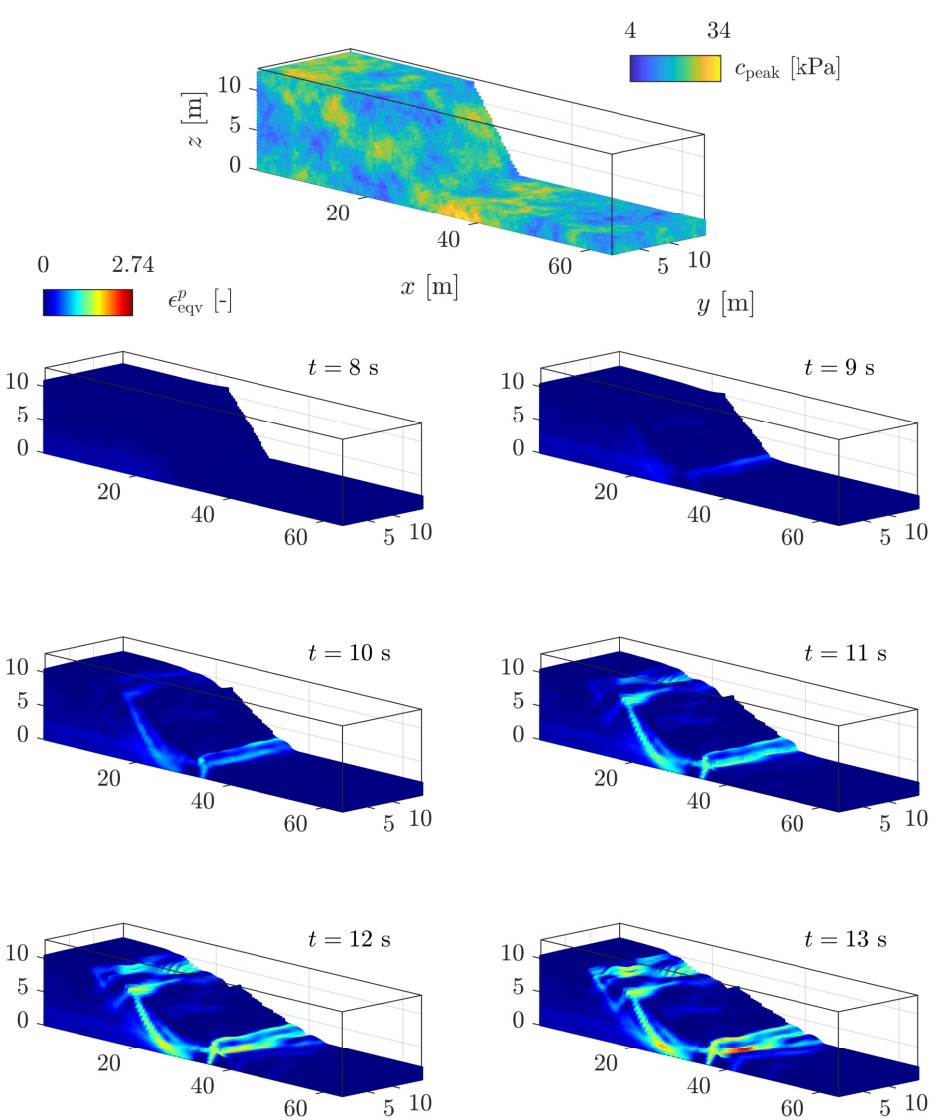

**Figure 16.** Heterogeneous cohesion field with an exponential covariance function: time evolution of the equivalent plastic strain $\epsilon_{\mathrm{eqv}}^p$. Similar to Fig. D2, heterogeneous behaviour is observed. However, the exponential covariance function results in an even more complex pattern of strain localization, i.e., minor and major scarps develop at the top. The crown-like structure of the slump becomes even more heterogeneous.



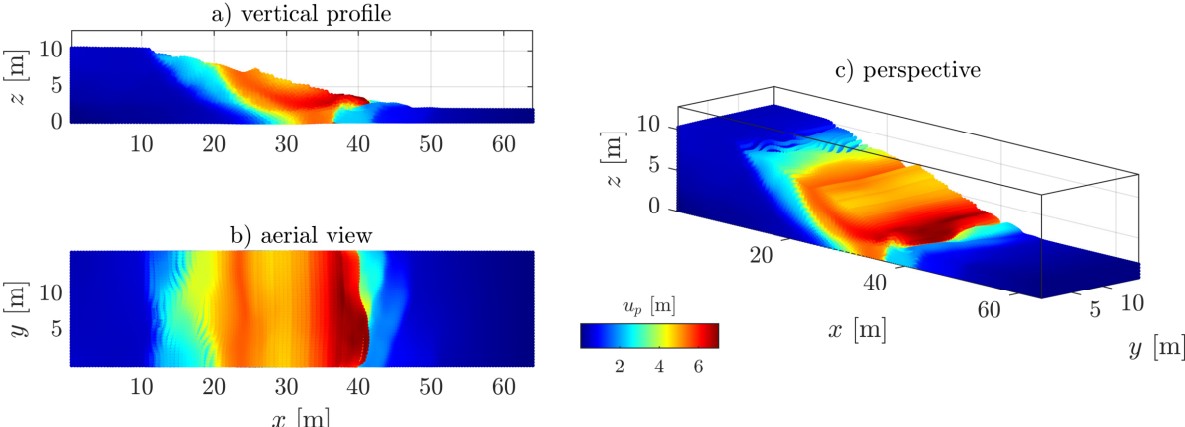

**Figure 17.** Displacement field obtained after $t = 15$ s for a heterogeneous peak cohesion field with an exponential covariance function.

et al. (2021), it is possible to overlap communications between multiple GPUs using asynchronous kernel executions. This allows us to achieve an optimal parallel efficiency: Räss et al. (2019a); Alkhimenkov et al. (2021) reported an efficiency of 95-98 % of the weak scaling tests involving up to 128 GPUs. Model 1b demonstrated the importance of the background mesh resolution over strain localization. Using a higher numerical resolution (i.e., finer background mesh) allows full resolution plastic strain localization. Similarly, future additional development efforts towards MPI implementation could resolve highly detailed three-dimensional granular collapse simulations in the future. This will definitely benefit future studies on complex strain localization.

Regarding the performance for Model 1b, the wall-clock time is $t_{GPU} = 1470.5$ s (25 min), and the number of iterations per second is 85.5 it·s$^{-1}$ for $n_{mp} = 819200$. As a preliminary example, the same numerical model was performed by Wyser et al. (2020a), who reported 19.98 it·s$^{-1}$ for $n_{mp} = 12800$. Proportionally, this corresponds to a performance gain factor of 275 for the GPU-based implementation (ep2-3De v1.0) over the MATLAB-based implementation (fMPMM-solver v1.1) (Wyser et al., 2020a).

### 5.3 Performance

The performance analysis we carried out in Model 2a demonstrated that even though the algorithm heavily relies on atomic operations to accumulate material point quantities on the nodes, the effective memory throughput reaches 88 % at most (for Tesla V100). We expected a much lower throughput due to the use of these atomic operations, since they are likely known to undermine the computational performances of an implementation under previous GPU architectures (e.g., Kepler) (Dong et al., 2015a; Dong and Grabe, 2018; Gao et al., 2018). Our actual understanding (at least for a GPU-based implementation of GIMPM) is that the latest GPU architecture (Ampere and Turing) is now efficient when dealing with atomic operations and that the need to use a complex data layout for scattering is not as important as before. Furthermore, we identify the



memory throughput as the main bottleneck: an additional 12 % performance improvement on the V100 before reaching the hardware limit of the memory bandwidth. The A100 shows that the solver reaches the hardware limit with an effective memory throughput which is very close (i.e., 97 %) to the actual maximum memory throughput. Similarly, the true limiting factor of our implementation is the hardware limit of the GPU on-chip memory. Further development efforts should be directed towards an MPI implementation of ep2-3De v1.0.

## 5.4 Slumping mechanics

We show the application of the GIMPM solver ep2-3De v1.0 for slumping mechanics. We have presented various slump results and demonstrated the significant influence of heterogeneities within the peak cohesion field over the displacement field or the equivalent plastic strain. However, we have arbitrarily selected values that resulted in severe deformations of the material, which we wanted to highlight to demonstrate the potential of the solver. Further efforts should now be oriented towards numerical models that are closer to real and well-documented cases, such as in Tran and Sołowski (2019); Ying et al. (2021). Despite the simplifications we made, we have reported three-dimensional simulations that resolve all the first-order characteristics of slumps, including complex major and minor scarps, different shear zones of various activities and a complex arrangement within the compression zone. The use of three-dimensional GIMPM implementation under a GPU architecture will highly benefit future studies in the field, allowing faster and detailed numerical simulations of heterogeneous and complex strain localization problems.

## 5.5 Code portability

Our numerical models showed the efficient computing capabilities of modern GPUs under the latest Nvidia GPU architectures. An important concern is the code portability. CUDA C is only applicable for Nvidia's GPUs and is not yet compatible with other corporation's GPUs, such as AMD (ATI Technologies). As such, an extension of the ep2-3De v1.0 solver towards an OpenCL-based implementation would ensure better code portability in the future.

## 6 Conclusions

We developed ep2-3De v1.0, an explicit GPU-based implementation of the generalized interpolation material point method that exploits the capabilities of the most recent GPU architectures (Ampere, Turing and Volta). We achieved fast execution times on a single GPU with a scattering approach that relies on extensive usage of atomic operations. We report, at most, an effective memory bandwidth of 88 % relative to the maximal hardware capabilities of the GPUs. We achieve, at most, a performance gain of 200× compared to a single-threaded CPU-based implementation of the solver. On entry-level customer electronics GPUs, we report a performance gain of ≈ 10×. We also report that the memory bandwidth is the main limiting performance factor. We validated our solver against the well-known experimental results of the granular collapse problem in a three-dimensional configuration. Furthermore, we show applications of the solver to model slumping mechanics considering different material heterogeneities.





Recent GPU architectures (Ampere, Turing and Volta) have certainly been optimized by Nvidia, increasing the efficiency of native atomic operations (i.e., scattering), as was suggested before by previous studies. This is encouraging, and future implementations on GPUs might be more straightforward and could now focus on using atomic operations instead of complex parallel implementations to avoid race conditions between threads without suffering from a dramatic loss of computational performance.

The single GPU implementation we propose here should be completed in the future by taking advantage of the message passing interface (MPI) and the computing power of a multi-GPU implementation to overcome the other limiting factor of ep2-3De v1.0, the GPU on-chip memory. Having such an implementation ensures that one can use a computationally powerful explicit GIMP solver to address complex elastoplastic problems in the field of geomechanics.

*Code availability.* The solver ep2-3De v1.0 developed in this study is licenced under the GPLv3 free software licence. The solver ep2-3De
v1.0$^2$ archive (v1.0) is available from a permanent DOI repository (Zenodo) at: https://doi.org/10.5281/zenodo.4966590 (Wyser et al., 2021).

---

$^2$The latest version of the code is available for download from Bitbucket at: https://bitbucket.org/ewyser/ep2-3de/src/master/ (last access: June 16, 2021).



## Appendix A: GIMPM basis functions and derivatives

One of the most important problems of any sMPM formulation is the cell-crossing instability (or error, see Steffen et al. 2008; Wilson et al. 2021). As material points move through the mesh, they cross element boundaries. The discontinuous gradient due to the $C_0$ continuity of the basis functions results in spurious oscillations of the stress field and internal forces (González Acosta et al., 2020, 2019; Bardenhagen and Kober, 2004) when material points cross element boundaries.

To solve for this instability, Bardenhagen and Kober (2004) introduced the generalized interpolation material point method (GIMPM). Whereas the material point is treated as a point in sMPM, Bardenhagen and Kober (2004) assigned a *spatial extent* or a *domain* to the material point. Alternative basis functions are constructed, i.e., to consider the material point domain, as follows:

$$\phi_{np} \equiv \phi_n(\boldsymbol{x}_p) = \frac{1}{v_p} \int_{\Omega_p \subset \Omega} \chi_p(\boldsymbol{x}) N_n(\boldsymbol{x}) \mathrm{d}\Omega, \tag{A1}$$

where $v_p$ is the material point volume, $\Omega_p$ denotes the material point domain, $\chi_p(\boldsymbol{x})$ is the *particle characteristic function*, $N_n(\boldsymbol{x})$ is the basis function (or shape function) for the mapping between the material point $p$ and its associated nodes $n$, and $\boldsymbol{x} = \boldsymbol{x}_p - \boldsymbol{x}_n$ are the local coordinates between node $n$ and material point $p$.

The particle characteristic function must satisfy the partition of unity property, i.e., $\sum_p \chi_p(\boldsymbol{x}) = 1$ (Bardenhagen and Kober, 2004). The simplest particle characteristic function is given by the hat function, i.e.,

$$\chi_p(\boldsymbol{x}) = \begin{cases} 1, & \text{if } \boldsymbol{x} \subset \Omega_p, \\ 0 & \text{otherwise.} \end{cases} \tag{A2}$$

The GIMPM basis functions and derivatives are constructed analytically (Coombs et al., 2020; Charlton et al., 2017) in one dimension from a convolution of the standard finite element basis functions and the material point characteristic function (Steffen et al., 2008), i.e.,

$$\phi_n(x_p) = \begin{cases} 1 - (4x_p^2 + l_p^2)/(4hl_p) & \text{if } |x_p| < l_p/2 \\ 1 - |x_p|/h & \text{if } l_p/2 \leq |x_p| < h - l_p/2 \\ (h + l_p/2 - |x_p|)^2/(2hl_p) & \text{if } h - l_p/2 \leq |x_p| < h + l_p/2 \\ 0 & \text{otherwise }, \end{cases} \tag{A3}$$

where $l_p$ is the length of the material point domain, $h$ is the mesh resolution, and $x = x_p - x_n$, where $x_p$ is the coordinate of a material point and $x_n$ is the coordinate of its associated node $n$. The two-dimensional basis function of a node $n$ with its material point $p$ is constructed as

$$\phi_{np} \equiv \phi_n(\boldsymbol{x}_p) = \phi_n(x_p)\phi_n(y_p), \tag{A4}$$

for which the gradient is defined as

$$\nabla \phi_{np} \equiv \nabla \phi_n(\boldsymbol{x}_p) = (\partial_x \phi_n(x_p)\phi_n(y_p), \phi_n(x_p)\partial_y \phi_n(y_p)). \tag{A5}$$



## Appendix B:  Gaussian random cohesion fields

In Earth Sciences, random fields (Christakos, 1992) are numerically generated predictions of a geophysical property (i.e., rock- or soil-related properties) with probabilistic spatial variability. These predictions are based on i) an assumed probability density function, i.e., characterized by a mean value $\mu$ with a standard deviation $\sigma$, and ii) an assumed spatial correlation function, characterised by fluctuation scales in a vector format, i.e., $\boldsymbol{\lambda} = (\lambda_x, \lambda_y, \lambda_z)$. In regard to numerical modelling, the principal requirement is that both small and large scales are simultaneously resolved over the computational mesh to ensure physically meaningful solutions.

Recently, Räss et al. (2019b) presented an efficient implementation based on a spectral representation of Gaussian random fields for geophysical applications using either Gaussian or exponential covariance functions. The numerical codes, named GRFS, were made available by Räss et al. (2019b) in both native MATLAB and CUDA C languages [3]. However, a sufficiently large number of harmonics should be used to obtain convergent Gaussian random fields, as stated in Räss et al. (2019b).

Similar to the random material point method (RMPM, see Wang et al. 2016a; Liu et al. 2019; Remmerswaal et al. 2021) initially proposed by Fenton and Vanmarcke (1990) to generate RFs for a finite element mesh (RFEM), we combined this approach with the codes proposed by Räss et al. (2019b) to generate an isotropic peak cohesion field to demonstrate its influence on the mechanical behaviour.

## Appendix C:  Volumetric locking and damping corrections

In Huang et al. (2015), no volumetric locking mitigation strategy was introduced, even though tough low-order elements were used. This should result in severe volumetric locking issues and an overall stiffer response of the granular material. In addition, Huang et al. (2015) used the standard (or original) material point method (instead of the generalized interpolation material point method), which is well known to introduce spurious oscillations of internal forces (González Acosta et al., 2020).

When implementing the proposed volumetric locking mitigation strategy, we observed a) larger deformations of the granular material with a stronger vertical compaction (i.e., stronger vertical displacement) and b) slightly longer run-out distances when compared to the experimental data. The softer mechanical response of the granular material had to be compensated somehow, which can be achieved by the introduction of a small local damping parameter.

We reproduced the numerical setting used in Huang et al. (2015) with the same mesh resolution, i.e., $\Delta x = \Delta y = 2.5$ mm, and a similar number of material points $n_{mp} = 28800$ with an initial number of material points per initially filled element $n_{pe} = 9$. The material parameters used for this preliminary investigation are presented in §4.1.1.

Figure C1 a) & b) shows the major differences between either a locking-free or a locking-prone solution and the experimental results. As mentioned before, a slightly longer run-out distance is obtained for the locking-free solution. As a result, the numerical prediction given by the locking-free solution of the free surface is underestimated. However, the most noticeable difference is the failure surface. Whereas the failure surface predicted by the locking-prone solution fits with the experiment of

---

[3] available at TheroutinesGRFSareavailableathttps://bitbucket.org/lraess/grfs/src/master/

Bui et al. (2008), it diverges for a locking-free solution. In particular, the onset of the failure surface at the top of the material is underestimated by the locking-free solution compared to the experimental results. This is due to the softer response of the granular material when volumetric locking is mitigated, which promotes greater vertical compaction and stronger run-out distance at the same time.

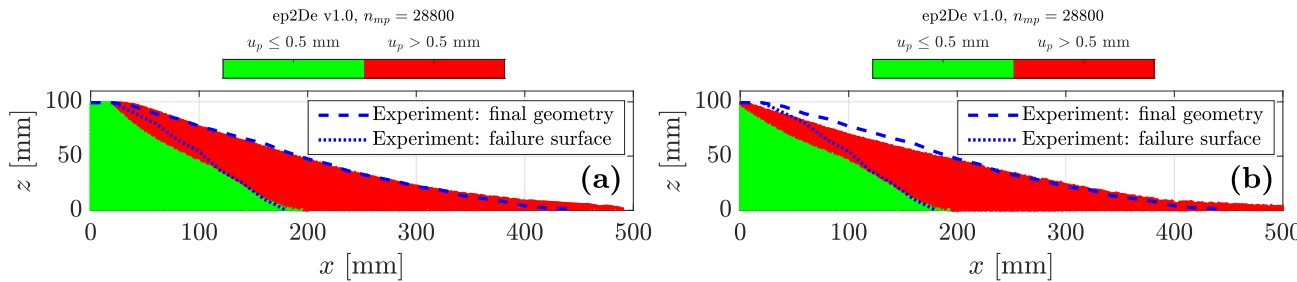

**Figure C1.** a) Numerical solution without any volumetric locking strategy and b) numerical solution with the proposed volumetric locking strategy. For both cases, no damping is introduced.

Even though the introduction of local damping better resolves the experimental results, one can argue that the locking-free solution without the introduction of local damping still agrees with the experiment of Bui et al. (2008). The overall response of the numerical granular collapse is still very close to the actual physical experiment, and the differences between the numerical and experimental results can still be considered acceptable.





## Appendix D: Heterogeneities for the peak cohesion field

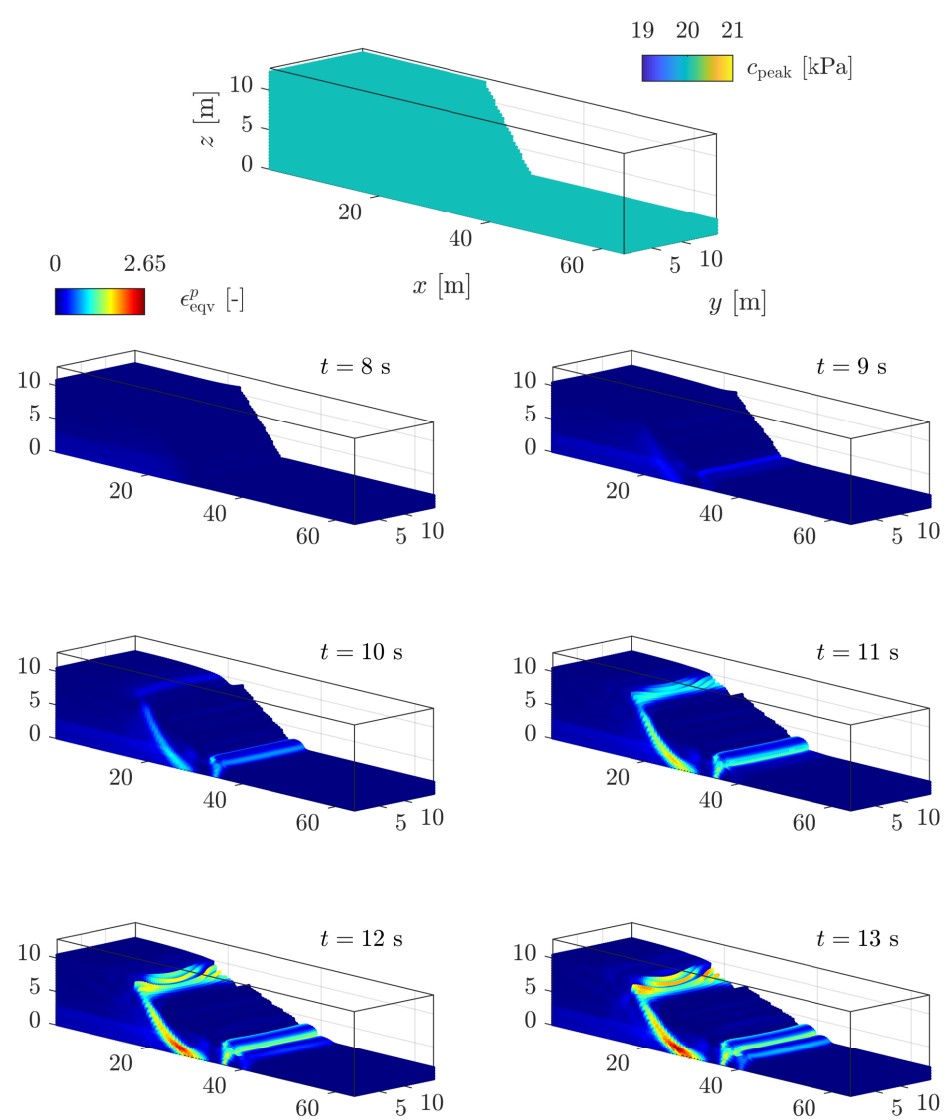

**Figure D1.** Homogeneous cohesion field: time evolution of the equivalent plastic strain $\epsilon_{eqv}^p$. Its evolution is rather homogeneous, and the overall plastic behaviour is free of any heterogeneities. Some of the first-order characteristics are observed, i.e., a principal shear zone and a compression zone at the toe of the slump.



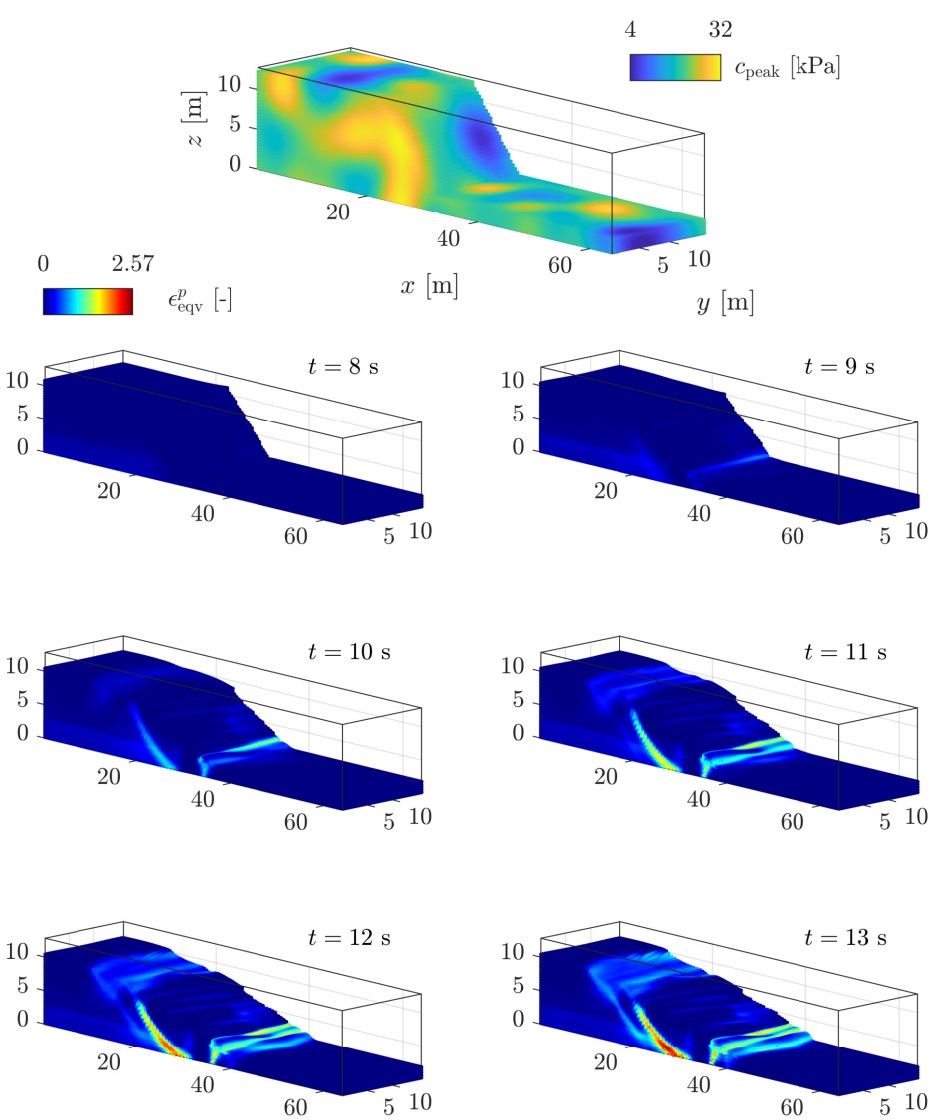

**Figure D2.** Heterogeneous cohesion field with a Gaussian covariance function: time evolution of the equivalent plastic strain $\epsilon^p_{\mathrm{eqv}}$. Unlike Fig. D1, heterogeneous behaviour is observed, i.e., the appearance of a second shear zone highlights a more complex deformation pattern. Moreover, a crown-like structure starts to develop at the top of the material, where an initial weak zone is located.

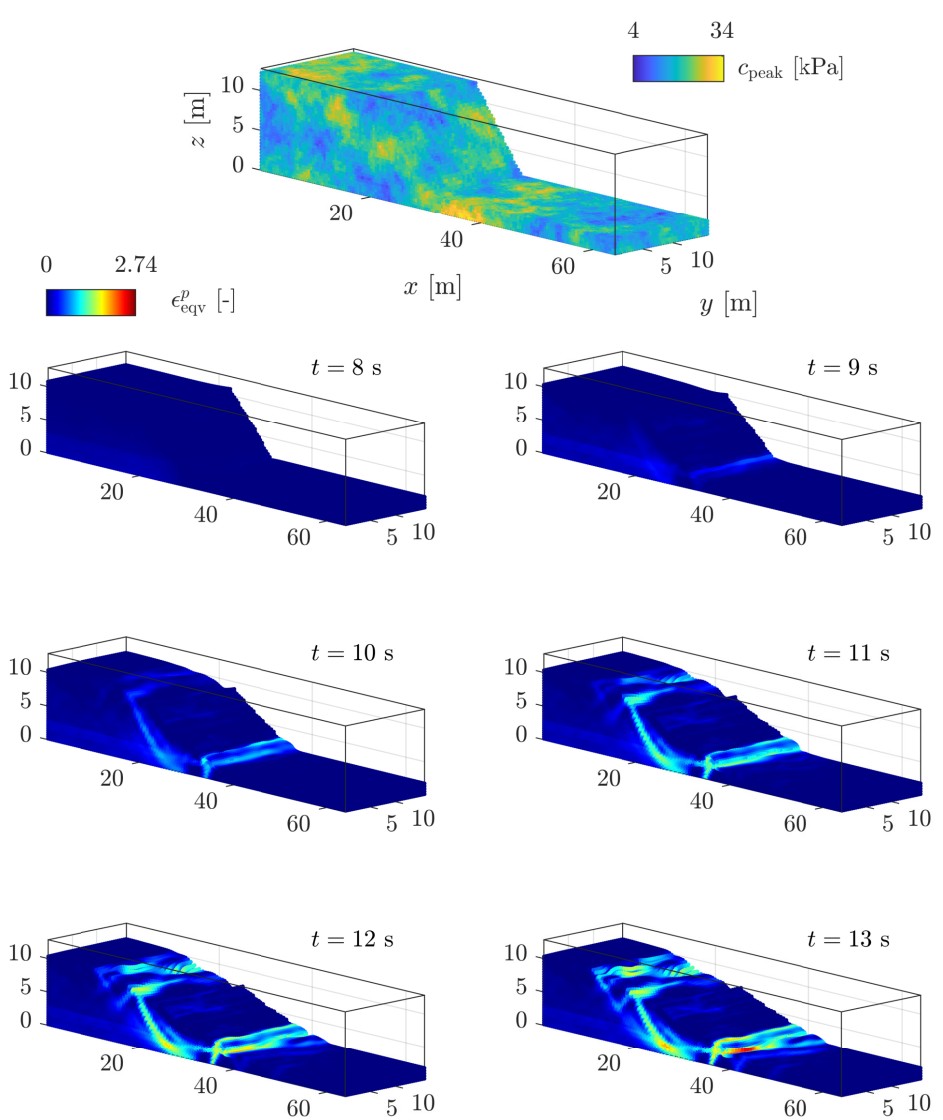

**Figure D3.** Heterogeneous cohesion field with an exponential covariance function: time evolution of the equivalent plastic strain $\epsilon^p_{\mathrm{eqv}}$. Similar to Fig. D2, heterogeneous behaviour is observed. However, the exponential covariance function results in an even more complex pattern of strain localization, i.e., minor and major scarps develop at the top. The crown-like structure of the slump becomes even more heterogeneous.



*Author contributions.* EW and YA wrote the original manuscript and developed the first version the ep2-3De v1.0 solver. MJ and YP supervised the early stages of the study and provided guidance. All authors have reviewed and approved the final version of the paper.

*Competing interests.* The authors declare that they have no conflicts of interest.

*Acknowledgements.* Yury Alkhimenkov gratefully acknowledges support from the Swiss National Science Foundation (grant no. 172691). Yury Alkhimenkov and Yury Y. Podladchikov gratefully acknowledge support from the Russian Ministry of Science and Higher Education (project No. 075-15-2019-1890).



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
