# Peer review of "An explicit GPU-based material point method solver for elastoplastic problems (ep2-3De v1.0)"

_Geoscientific Model Development, 2021_

## Author Comment (AC1)

**Author responses to the Interactive discussion on "An explicit GPU-based material point method solver for elastoplastic problems (ep2-3De v1.0)" in the Geoscientific Model Development (GMD) Journal**

Emmanuel Wyser, Yury Alkhimenkov, Michel Jaboyedoff, Yury Podladchikov

August 28, 2021

The referee comments appear in black, whereas our responses appear in blue and the changes made in the revised manuscript appear in red.

**1 Reviewer #1**

The authors implemented an explicit GPU-based solver within the material point method framework and tested using two- and three-dimensional problems. Results seem to agree with the expected values validating this MPM - GPU architecture. I would like to suggest the publication of this work. Nonetheless, some minor points should be addressed first.

We would like to acknowledge the reviewer for the time spent on the revision of our work.

**Comment # 1** The authors mentioned that this GPU architecture speeds up information transfer between nodes and material points. As stated, this is one of the most computationally expensive operations in MPM. Nevertheless, finding material points new location after the mesh returns to its original position is another process that is computationally expensive (in many cases more expensive than information transfer between nodes and material points). I would like to know if the GPU architecture proposed also improves this step. If yes, the author could indicate it in the paper. If not, it would be interesting if the author discusses the possibility of combining some techniques (e.g. Pruijn N.S. 2016) together with GPU's to improve MPM computations.

Pruijn N.S. 2016. The improvement of the material point method by increasing efficiency and accuracy. TU Delft Master Thesis.

**Reply # 1** The reviewer points out the computationally expansive operation of finding material point's new locations after the mesh has been reset at the end of a time step. We used a regular background mesh (as opposed to triangular mesh or non-constant element size), therefore, it is straightforward to find material point's new location. To find in which element $e$ a material point $p$ is located, we use the following equation

$$e = (\texttt{floor}((z_p - z_{\min})/\Delta z) + n_{el,z} \times \texttt{floor}((x_p - x_{\min})/\Delta x)) + n_{el,x} \times n_{el,z}\texttt{floor}((y_p - y_{\min})/\Delta y), \quad (1)$$

where $n_{el,x}$ and $n_{el,z}$ are the number of elements along $x-$ and $z-$directions, $x_{\min}$, $y_{\min}$ and $z_{\min}$ are the minimum $x$, $y$, and $z$ coordinate of nodes. However, this is far less trivial when irregular background mesh is used and such equation can not be used any more. Such concern is out of the scope of our contribution since our implementation only consider a regular background mesh. This concern also explain why we selected a regular background mesh.

We looked at the reference suggested by the reviewer. From an overlook of the mentioned research work, we think a GPU-based implementation of the brute force method is possible, but this would require deeper

investigations. This could be the subject of future studies. We strongly think such method could benefit from the computational power of GPUs.

**Change # 1** -

**Comment # 2** The authors include a damping value $D$ in the simulations. This value is not well validated. It seems that several simulations were needed to find it, giving the idea that $D$ is not related to the material properties and geometry and is more of an artificial way to reach the desirable results. The authors should elaborate better on the reasons for using this specific damping value.

**Reply # 2** It is true that the damping value is not well validated. From a broader perspective and to our knowledge, no studies thoroughly investigated and quantified the influence of damping. However, the common range between 0.05 and 0.15 is usually selected by researchers (e.g., Wang et al., 2016b; Wang et al., 2016a) for dynamic analysis. This range was found sufficient to damp out dynamic oscillations while not producing spurious plastic yielding or an over-damped system, as mentioned in Wang et al., 2016a. As such, we decided to use a damping value of $D = 0.1$, since reasonable propagations were obtained and no spurious plastic yielding were noticed.

As raised by the reviewer, we will elaborate better on the reasons of this specific value and will clarify accordingly the lack of validation of this damping value within the main body of the text. We will also mention the need for future studies addressing this concern.

**Change # 2** -

**Comment # 3** In line 32, the authors mentioned that "The background mesh can be reset". As far as I know, the background mesh must be reset. I recommend changing the verb "can" for a better one.

**Reply # 3** We agree with the reviewer. We will change the verb in the revised manuscript.

**Change # 3** L.32 The background mesh is reset

**Comment # 4** I am wondering if the variables in line 75 are the same as in equations 2 and 3 since different punctuations were used (e.g. Å« and û).

**Reply # 4** We recognize here that this is a mathematical typo. Variables in line 75 are the same as in Eqs. 2 and 3. We will fix this in the revised version of the manuscript.

**Change # 4** (L.75) $\hat{u} \rightarrow \bar{u}$ and $\hat{\tau} \rightarrow \bar{\tau}$

**Comment # 5** Finally, I recommend reading the paper again to correct some typos detected..

**Reply # 5** We acknowledge the reviewer's recommendation and we will read the manuscript again to correct remaining typos

**Change # 5** -

**Reply # 6**   We additionally extended the original single GPU code and implemented a multi-GPU version using the message passing interface standard. We provide the new section below, which then will be included (with some simplifications) into the manuscripts during the revision stage.

**The Multi-GPU Code Implementation**

**Introduction**

One of the major limitation of `ep2-3De v1.0` is the on-chip memory. We demonstrated that an implementation of the material point framework quickly reaches the hardware limit of GPUs, even on modern architectures. It is then essential to overcome this limit in order to resolve larger computational domain with a greater amount of material points.

   Here, we address this concern by implementing a distributed memory parallelisation using the message passing interface (MPI) standard. However, we limit our implementation efforts by considering 1) a one-dimensional GPU topology, 2) no computation/communication overlaps, and 3) only mesh-related quantities are shared amongst GPUs, i.e., the material points are not transferred between GPUs during a simulation. We also selected a non-adaptive time step to avoid the collection of the material point's velocities located in different GPUs at the beginning of each calculation cycle.

**Available computational resources**

The multi-GPU simulations are performed on the supercomputer Octopus running on a CentOS with the latest CUDA version v11. The multi-GPU simulations are run on the two different systems. The first one is an Nvidia DGX-1 - like node hosting 8 Nvidia Tesla V100 Nvlink (32 GB) GPUs, 2 Intel Xeon Silver 4112 (2.6 GHz) CPUs. The second one is composed of 32 nodes, each featuring 4 Nvidia GeForce GTX Titan X Maxwell (12 GB) GPUs, 2 Intel XEON E5-2620V3 4112 (2.4 GHz) CPUs. To summarize the computational resources in use, Table 1 presents the main characteristics of the GPUs used in this study.

Table 1: List of the graphical processing units (GPUs) used for multi-GPU simulations.

| GPU | Architecture | On-chip memory [GB] |
|---|---|---|
| $8\times$V100 | Volta | $8\times32$ |
| $128\times$GTX Titan X | Maxwell | $128\times12$ |

**Model 2a**

To avoid frequent material point's transfers amongst the GPUs, we consider an overlap of 8 elements between neighbouring meshes, i.e., 9 nodes. This results in a one-dimensional GPU topology, for which both material points and meshes are distributed along the $y-$direction of the global computational domain (see Figs. 1 & 2). Arranging GPUs along this direction allows to overcome the need to transfer material points amongst GPUs, provided that the material point's displacement is not greater than the buffer zone, i.e., the element overlap. The evaluation of the multi-GPU implementation is based on the Model 2a, with slight modifications, i.e., the number of element along the $y$-direction is largely increased. The size of the physical domain $l_z \times l_x \times l_y$ is, at most, 12 m $\times$ 64 m $\times$ (64$\times$2048) m.

**Model 2a: multi-GPU performances**

We consider two distributed computing systems for parallel GPU computation, using up to 8 Tesla V100 (Volta architecture) or 128 Geforce GTX Titan X (Maxwell architecture). All numerical simulations are performed using a single-arithmetic precision (i.e., $n_\mathrm{p} = 4$ bytes). This allows to increase the maximum number of material points and mesh dimensions. In addition, our GPU implementation relies on the usage of the built-in function `atomicAdd()`. It does not support the double-precision floating-point format FP64 for GPUs with compute capabilities lower than 6.0, i.e., the Maxwell architecture amongst others.

[Figure]

Figure 1: Geometry for the earth slump. For the multi-GPU implementation, the number of element along the $y-$direction can be largely increased, i.e., $n = 2048$.

[Figure]

Figure 2: Domain partition of the material points amongst 8 GPUs. Combined with an overlap of 8 elements along the $y$-direction, material points can moderately move while still residing within the same GPU during the whole simulation.

Note that, unlike the Tesla V100, the Geforce GTX Titan X only delivers an effective memory throughput of $\mathrm{MTP_{eff}} \approx 100 \ \mathrm{GB}s^{-1}$. This corresponds to 38 % of its hardware limit. This was already reported by Räss et al., 2019; Alkhimenkov et al., 2021 and, it could be attributed to its older Maxwell architecture (Gao et al., 2018). This performance drop is even more severe, mainly due to the use of built-in functions like `atomicAdd()`.

**Computing system: up to 8 Tesla V100**

We first performed parallel simulations with a moderate number of GPUs, i.e., up to 8 Tesla V100 NVlink (32 GB). The respective wall-clock times are reported in Fig. 3. We report a wall-clock time of $\approx 110$ s for $n_{mp} \approx 10^8$. For the same amount of material points, we report a roughly weak scaling between the number of GPUs and the wall-clock time. If $n_{mp}$ is increased by a factor 2, 4 or 8, the wall-clock time is roughly similar to the baseline, i.e., $n_{\mathrm{GPU}} = 1$.

Such weak scaling is more obvious when inspecting the $\mathrm{MTP_{eff}}$ measured (see Fig. 4), i.e., the total sum of $\mathrm{MTP_{eff}}$ across all the GPUs. Based on the memory throughput of 1 GPU, an estimation of a perfect weak scaling is possible. For 8 GPUs, it should correspond to $\mathrm{MTP_{eff}} = 4824 \ \mathrm{GB}s^{-1}$, whereas we report $\mathrm{MTP_{eff}} = 4538 \ \mathrm{GB}s^{-1}$. This gives a parallel efficiency of $\approx 94\%$ and, an effective speed-up of $7.5\times$. Similar observations are made for $n_{\mathrm{GPU}} = 2$ and $n_{\mathrm{GPU}} = 4$.

**Computing system: up to 128 Geforce GTX Titan X**

We investigate a parallel GPU computing using up to 128 Geforce GTX Titan X. This allows to address even larger geometries, as showed in Fig 5 where a geometry of nearly $n_{mp} \approx 8 \cdot 10^8$ is resolved in less than 8

[Figure]

Figure 3: Wall-clock time for 1, 2, 4 and 8 Tesla V100 GPUs.

[Figure]

Figure 4: Sum across the GPUs involved of the $\text{MTP}_{\text{eff}}$. We roughly report a weak scaling between the number of GPUs and the overall effective memory throughput.

minutes. The first observation is that, for parallel computing up to 64 GPUs, the wall-clock time evolution is smooth. For 128 GPUs, the wall-clock time is chaotic for fewer material points whereas it stabilizes as the number of material points increases. We suspect the absence of computation/communication overlaps to be the main reason of this erratic behaviour. The communication between many GPUs requires careful synchronization between GPUs which can be hidden under computation/communication overlap. The total size of the overlap is constant, regardless of the $y-$dimension. As the number of material points increases, the time spent on computation becomes larger compared to the time spent on exchanges between GPUs and the wall-clock time stabilizes.

[Figure]

Figure 5: Wall-clock time reported for up to 128 Geforce GTX Titan X GPUs and up to $n_{mp} \approx 8 \cdot 10^8$.

Another observation is the effective memory throughput (see Fig. 6). When considering a perfect weak scaling, one should measure an effective memory throughput $\text{MTP}_{\text{eff}} = 12800 \text{ GB}s^{-1}$ for 128 GPUs whereas we report only $\text{MTP}_{\text{eff}} = 10953 \text{ GB}s^{-1}$. This gives a parallel efficiency of $\approx 85\%$ and, an effective speed-up of $\approx 110\times$. When using less GPUs, the parallel efficiency is higher, i.e., 98 % for 8 GPUs.

[Figure]

Figure 6: MTP$_{\text{eff}}$ sum across the GPUs involved.

**Discussion**

Even tough the simplifications made alleviate the on-chip memory limitation, the type of problem, which can be addressed, is reduced. As an example, investigating high-resolution three-dimensional granular collapses is not possible under the assumptions made, because of small displacement required along the $y-$direction. This is incompatible with three-dimensional granular collapses. Hence, this motivates future deeper investigations toward a more versatile multi-GPU implementation. In addition, we report a slight drop of the parallel efficiency, as the number of GPUs increases. Future works should be directed toward a parallel strategy that hides communication latency, as proposed in Räss et al., 2019; Räss et al., 2020; Alkhimenkov et al., 2021.

However, such multi-GPU implementation is particularly well-suited to resolve highly-detailed three-dimensional shear-banding. We also reported decent wall-clock times (less than 8 minutes) for simulations with nearly a billion material points. One could argue that limiting the material point method to small displacement is a non-sense. Essentially, finite element codes are better suited for small strain analysis. However, this gives interesting insights on a multi-GPU implementation of the material point framework on a GPU supercomputer.

**References**

Alkhimenkov, Y., L. Räss, L. Khakimova, B. Quintal, and Y. Podladchikov (2021). "Resolving wave propagation in anisotropic poroelastic media using graphical processing units (GPUs)". In: *Journal of Geophysical Research: Solid Earth* n/a.n/a. e2020JB021175 2020JB021175, e2020JB021175. DOI: `https://doi.org/10.1029/2020JB021175`.

Gao, M., X. Wang, K. Wu, A. Pradhana, E. Sifakis, C. Yuksel, and C. Jiang (Dec. 2018). "GPU Optimization of Material Point Methods". In: *ACM Trans. Graph.* 37.6. DOI: `10.1145/3272127.3275044`.

Räss, L, T. Duretz, and Y. Podladchikov (2019). "Resolving hydromechanical coupling in two and three dimensions: spontaneous channelling of porous fluids owing to decompaction weakening". In: *Geophysical Journal International* 218.3, pp. 1591–1616.

Räss, L., A. Licul, F. Herman, Y. Y. Podladchikov, and J. Suckale (2020). "Modelling thermomechanical ice deformation using an implicit pseudo-transient method (FastICE v1. 0) based on graphical processing units (GPUs)". In: *Geoscientific Model Development* 13.3, pp. 955–976.

Wang, B., P. J. Vardon, and M. A. Hicks (2016a). "Investigation of retrogressive and progressive slope failure mechanisms using the material point method". In: *Computers and Geotechnics* 78, pp. 88–98. DOI: `https://doi.org/10.1016/j.compgeo.2016.04.016`.

Wang, B., P. J. Vardon, M. A. Hicks, and Z. Chen (2016b). "Development of an implicit material point method for geotechnical applications". In: *Computers and Geotechnics* 71, pp. 159–167. DOI: `https://doi.org/10.1016/j.compgeo.2015.08.008`.

---

## Author Response (AR1)

**Author responses to the Interactive discussion on "An explicit GPU-based material point method solver for elastoplastic problems (ep2-3De v1.0)" in the Geoscientific Model Development (GMD) Journal**

Emmanuel Wyser, Yury Alkhimenkov, Michel Jaboyedoff, Yury Podladchikov

October 28, 2021

The referee comments appear in black, whereas our responses appear in blue and the changes made in the revised manuscript appear in red.

**1 Reviewer #1**

The authors implemented an explicit GPU-based solver within the material point method framework and tested using two- and three-dimensional problems. Results seem to agree with the expected values validating this MPM - GPU architecture. I would like to suggest the publication of this work. Nonetheless, some minor points should be addressed first.

We would like to acknowledge the reviewer for the time spent on the revision of our work.

**Comment # 1** The authors mentioned that this GPU architecture speeds up information transfer between nodes and material points. As stated, this is one of the most computationally expensive operations in MPM. Nevertheless, finding material points new location after the mesh returns to its original position is another process that is computationally expensive (in many cases more expensive than information transfer between nodes and material points). I would like to know if the GPU architecture proposed also improves this step. If yes, the author could indicate it in the paper. If not, it would be interesting if the author discusses the possibility of combining some techniques (e.g. Pruijn N.S. 2016) together with GPU's to improve MPM computations.

Pruijn N.S. 2016. The improvement of the material point method by increasing efficiency and accuracy. TU Delft Master Thesis.

**Reply # 1** The reviewer points out the computationally expansive operation of finding material point's new locations after the mesh has been reset at the end of a time step. We used a regular background mesh (as opposed to triangular mesh or non-constant element size), therefore, it is straightforward to find material point's new location. To find in which element $e$ a material point $p$ is located, we use the following equation

$$e = (\texttt{floor}((z_p - z_{\min})/\Delta z) + n_{el,z} \times \texttt{floor}((x_p - x_{\min})/\Delta x)) + n_{el,x} \times n_{el,z}\texttt{floor}((y_p - y_{\min})/\Delta y), \quad (1)$$

where $n_{el,x}$ and $n_{el,z}$ are the number of elements along $x-$ and $z-$directions, $x_{\min}$, $y_{\min}$ and $z_{\min}$ are the minimum $x$, $y$, and $z$ coordinate of nodes. However, this is far less trivial when irregular background mesh is used and such equation can not be used any more. Such concern is out of the scope of our contribution since our implementation only consider a regular background mesh. This concern also explain why we selected a regular background mesh.

We looked at the reference suggested by the reviewer. From an overlook of the mentioned research work, we think a GPU-based implementation of the brute force method is possible, but this would require deeper

investigations. This could be the subject of future studies. We strongly think such method could benefit from the computational power of GPUs.

**Change # 1** -

**Comment # 2** The authors include a damping value $D$ in the simulations. This value is not well validated. It seems that several simulations were needed to find it, giving the idea that $D$ is not related to the material properties and geometry and is more of an artificial way to reach the desirable results. The authors should elaborate better on the reasons for using this specific damping value.

**Reply # 2** It is true that the damping value is not well validated. From a broader perspective and to our knowledge, no studies thoroughly investigated and quantified the influence of damping. However, the common range between 0.05 and 0.15 is usually selected by researchers (e.g., Wang et al., 2016b; Wang et al., 2016a) for dynamic analysis. This range was found sufficient to damp out dynamic oscillations while not producing spurious plastic yielding or an over-damped system, as mentioned in Wang et al., 2016a. As such, we decided to use a damping value of $D = 0.1$, since reasonable propagations were obtained and no spurious plastic yielding were noticed.

As raised by the reviewer, we will elaborate better on the reasons of this specific value and will clarify accordingly the lack of validation of this damping value within the main body of the text. We will also mention the need for future studies addressing this concern.

**Change # 2** We decided to include this concern in the Discussion section of the manuscript, see L.580-584.

**Comment # 3** In line 32, the authors mentioned that "The background mesh can be reset". As far as I know, the background mesh must be reset. I recommend changing the verb "can" for a better one.

**Reply # 3** We agree with the reviewer. We will change the verb in the revised manuscript.

**Change # 3** L.32 The background mesh is reset

**Comment # 4** I am wondering if the variables in line 75 are the same as in equations 2 and 3 since different punctuations were used (e.g. Å≪ and û).

**Reply # 4** We recognize here that this is a mathematical typo. Variables in line 75 are the same as in Eqs. 2 and 3. We will fix this in the revised version of the manuscript.

**Change # 4** (L.75) $\hat{u} \rightarrow \bar{u}$ and $\hat{\tau} \rightarrow \bar{\tau}$

**Comment # 5** Finally, I recommend reading the paper again to correct some typos detected..

**Reply # 5** We acknowledge the reviewer's recommendation and we will read the manuscript again to correct remaining typos

**Change # 5** -

**2 Reviewer #2**

This paper demonstrates the numerical implementation of the explicit material point method using GPU for parallel computing. In Reviewer opinion, the paper is of interest for the Readers of Geoscientific Model Development. However, a number of issues need to be addressed before the paper can be accepted for publication.

We would like to acknowledge the reviewer for the time spent on the revision of our work.

**Comment # 1** The paper is currently long and several part can be replaced by references. For example, the MPM algorithm is derived from forward-Euler scheme with update stress lass, GIMP basis functions in appendix A.

**Reply # 1** Thank you for raising that point. We recognize that the paper can be long for the audience. However, we also think it is the bare minimum to have a grasp over the material point method. We strongly believe this avoids the reading of numerous references and facilitate the overall understanding of the method to a broader audience. We already attempted to shorten the section 2 by putting in Appendix a detailed description of the shape functions used in GIMPM.

**Change # 1** -

**Comment # 2** There are several methods dealing with the volumetric locking in the literature. However, the author proposed the volumetric locking by averaging only the volumetric part of the stress tensor. The author is suggested to clarify the decision for that. Furthermore, volumetric locking can smooth out the value of the stress. It is better if the stress is plotted in the numerical examples to see the difference between simulations with and without volumetric locking.

**Reply # 2** It is true that in the literature, the stress tensor $\boldsymbol{\sigma}$ is averaged (Mast et al., 2012; Cuomo et al., 2019; Lei et al., 2020). Volumetric locking procedures generally mitigates locking by a more appropriate formulation of the volumetric component of the strain. The B-bar method ($\bar{B}$, see Bisht et al. 2021) splits the strain tensor into deviatoric and spherical parts. Deviatoric strains are evaluated at the material point's coordinate whereas the spherical part is evaluated only at the element center's coordinate. This yield a stress tensor, for which the pressure is defined at the element's center.

As such, we modified the element-based strategy (Mast et al., 2012; Cuomo et al., 2019; Lei et al., 2020), considering only the pressure term, which need to be averaged at the element's center. We believe this formulation is more consistent with the essence of the B-bar method, which splits the deviatoric part from the spherical part.

It is true that volumetric locking can smooth out the value of stresses. As suggested, we will add plots to show the difference between a locking-free and a locking-prone solution in the revised manuscript in the Appendix C: Volumetric locking and damping corrections.

We here show an example of slumping considering the geometry as in Model 2b. We selected an homogeneous initial cohesion field with values presented in the submitted manuscript. Figure 1 demonstrates that a significantly smoother pressure field is resolved with the proposed method. In addition, the pressure field is smoothed but it does not significantly differs from the original pressure field (in locations where locking is minimum). Volumetric locking is particularly highlighted within shear bands due to isochoring plastic flows, resulting in significant stress oscillations.

**Change # 2.1** (L.274) We believe this approach is conceptually closer to the B-bar technique.

**Change # 2.2** (Appendix C: Volumetric locking and damping corrections) We added new figures to show the difference between a locking-free and locking-prone solution.

[Figure]

(a) Non-smooth pressure field due to volumetric locking

[Figure]

(b) Smooth pressure field when volumetric locking is mitigated with the proposed solution

Figure 1: Element-based reconstruction of the pressure field to mitigate volumetric locking issues, with a total number of material points $n_{mp} \approx 10^5$.

**Comment # 3**  Section 4 mentions that Model 1b demonstrates the influence of mesh resolution but I do not see it in Model1b. The author is suggested to perform convergence rate analysis in different mesh size in the plane strain to highlight the influence of the mesh resolution.

**Reply # 3**  We understand the concern of the reviewer. It is true that the mesh resolution is not clearly demonstrated in Model1b under plane strain conditions.

We here present additional plots (see Fig. 2) to show to the reviewer the influence of the mesh resolution over shear banding. All the parameters used are the same as described in the original manuscript, except that the number of element in the $x$-direction is increased, *i.e.,* $n_{el,x} = 40, 80, 160, 320$. We can clearly observe that as the mesh resolution increases, the shear band resolution gets more accurate, *i.e.,* the finer the resolution the finer the shear band thickness. We also observe a more complex shear banding pattern at the bottom right, *e.g.,* $x > 140$ mm and $y < 50$ mm. Again, such shear banding arrangement is better resolved with a finer mesh resolution. We believe Fig. 2 clearly demonstrates the influence of the mesh resolution. Therefore, we do not think a convergence analysis is further needed.

**Change # 3**  We provide an additional sub-figure (see Figure 10). Figure 10 now demonstrates graphically the influence of the mesh resolution.

**Comment # 4**  For Model 1b, the presented final geometry of the experiment is shorter to the one in Bui et al. (2008) experiment (see Figure 6) in my opinion. Please check. Therefore, it is not necessary to introduce the damping.

[Figure]

Figure 2: Equivalent plastic strain for a variety of mesh resolution. One can see the influence of the mesh resolution over shear banding.

**Reply # 4**   Thank you for this remark. We checked and the reviewer is right. We performed additional simulations and still noticed that simulations without damping leaded to higher run-outs and softer response of the material. We will clarify this in the revised version of the manuscript.

**Change # 4**   We corrected Figures 6 and 9.

**Comment # 5**   In Model 2, there is a boundary effect on the failure mechanism as the shear band can touch the bottom boundary. It would be better if there is a larger depth in the bottom direction. And, the Model 2 introduces local damping which in my opinion it is not necessary.

**Reply # 5**   We acknowledge the boundary effect on the failure mechanism. However, our concern was to show an example of large deformation. A larger depth would also significantly reduces the total amount of deformation. Model 2 shows that several elasto-plastic features are resolved, such as bulging or thrusting at the toe of the slope. Such features would not be as obvious when a larger depth is considered. Concerning the introduction of a local damping, it is mainly to prevent an excessive run-out of the material when considering free-slip boundary conditions at the bottom of the computational domain.

However, we show here an example of a similar setting while considering a larger depth, as suggested by the reviewer. We briefly describe parameters that needed to be changed regarding the increase of the depth.

Regarding the initial geometry, we increase by a factor of two the $z$-direction and the $y-$direction. To avoid a significant elastic compaction of the material, we selected a higher Young's modulus, i.e., $E = 10$ MPa. This reduces the amount of vertical elastic compaction of the material during the elastic loading stage. No-slip boundary conditions are enforced at the base of the material. As thought by the reviewer, the local damping can be reduced. But a small value is still needed to suppress elastic wave propagation. In this case, we selected a damping value $D = 0.025$. An isotropic Gaussian random field is still selected for the cohesion field, with the same parameters used for Model 2b in the submitted manuscript.

Figure 3 shows the total displacement field after plastic yielding. In comparison with results in Model 2b, the magnitude of deformation is smaller. For instance, the intense bulging reported in the original manuscript is less evident in this setting. However, we still report thrusting mechanisms at the toe of the slope. Heterogeneous displacements are still observed, more evidently at the toe.

We also report a principal shear band and a heterogeneous crown-like structure (see Fig. 4). Plastic strain localisation differs regarding what was reported in the original manuscript. However, other simulations revealed that multiple shear bands can initiate successively. In Fig. 5, we can observe both shallower and deeper shear bands.

However, the deeper shear band is more developed than the shallower one. Similarly, we also report a more significant heterogeneous displacement field and a deeper thrusting mechanism at the toe of the slope.

[Figure]

Figure 3: Total displacement after elasto-plastic loading.

This is because of the random generation of the cohesion field, which can results in weaker and/or stronger local cohesion values. This also demonstrates an important influence of the initial cohesion field over the elasto-plastic response of the material.

[Figure]

Figure 4: Equivalent plastic strain after elasto-plastic loading.

Still, we think that the original results presented in the manuscript better represent the different mechanisms during the elasto-plastic deformation of an unstable material. It is true that increasing the depth greatly reduce the influence over the shear band propagation. We will clarify and discuss these considerations in the revised manuscript. We will also notify the reader that the boundary influence the propagation of the shear band due to the shallower depth mentioned by the reviewer.

**Change # 5**    (L.413-414) However, the bottom boundary condition influences the shear band propagation and the overall behaviour by introducing a stronger horizontal component in the motion.

[Figure]

[Figure]

Figure 5: Total displacement (up) and equivalent plastic strain (down) after elasto-plastic loading.

**Author comment # 6**   We additionally extended the original single GPU code and implemented a multi-GPU version using the message passing interface standard. We provide the new section below, which then will be included (with some simplifications) into the manuscripts during the revision stage.

**The Multi-GPU Code Implementation**

**Introduction**

One of the major limitation of `ep2-3De v1.0` is the on-chip memory. We demonstrated that an implementation of the material point framework quickly reaches the hardware limit of GPUs, even on modern architectures. It is then essential to overcome this limit in order to resolve larger computational domain with a greater amount of material points.

Here, we address this concern by implementing a distributed memory parallelisation using the message passing interface (MPI) standard. However, we limit our implementation efforts by considering 1) a one-dimensional GPU topology, 2) no computation/communication overlaps, and 3) only mesh-related quantities are shared amongst GPUs, i.e., the material points are not transferred between GPUs during a simulation. We also selected a non-adaptative time step to avoid the collection of the material point's velocities located in different GPUs at the beginning of each calculation cycle.

**Available computational resources**

The multi-GPU simulations are performed on the supercomputer Octopus running on a CentOS with the latest CUDA version v11. The multi-GPU simulations are run on the two different systems. The first one is an Nvidia DGX-1 - like node hosting 8 Nvidia Tesla V100 Nvlink (32 GB) GPUs, 2 Intel Xeon Silver 4112 (2.6 GHz) CPUs. The second one is composed of 32 nodes, each featuring 4 Nvidia GeForce GTX Titan X Maxwell (12 GB) GPUs, 2 Intel XEON E5-2620V3 4112 (2.4 GHz) CPUs. To summarize the computational resources in use, Table 1 presents the main characteristics of the GPUs used in this study.

Table 1: List of the graphical processing units (GPUs) used for multi-GPU simulations.

| GPU | Architecture | On-chip memory [GB] |
|---|---|---|
| $8\times$V100 | Volta | $8\times32$ |
| $128\times$GTX Titan X | Maxwell | $128\times12$ |

**Model 2a**

To avoid frequent material point's transfers amongst the GPUs, we consider an overlap of 8 elements between neighbouring meshes, i.e., 9 nodes. This results in a one-dimensional GPU topology, for which both material points and meshes are distributed along the $y-$direction of the global computational domain (see Figs. 6 & 7). Arranging GPUs along this direction allows to overcome the need to transfer material points amongst GPUs, provided that the material point's displacement is not greater than the buffer zone, i.e., the element overlap. The evaluation of the multi-GPU implementation is based on the Model 2a, with slight modifications, i.e., the number of element along the $y$-direction is largely increased. The size of the physical domain $l_z \times l_x \times l_y$ is, at most, 12 m $\times$ 64 m $\times$ (64$\times$2048) m.

[Figure]

Figure 6: Geometry for the earth slump. For the multi-GPU implementation, the number of element along the $y-$direction can be largely increased, i.e., $n = 2048$.

**Model 2a: multi-GPU performances**

[revised manuscript text omitted]

**Change # 6**  We introduced in multiple places in the revised manuscript the multi-GPU implementation presented above. Track changes in the revised version should be obvious. Therefore, we do not detail here every changes across the revised manuscript.

---

## Author Response (AR2)

**Author responses to the Interactive discussion on "An explicit GPU-based material point method solver for elastoplastic problems (ep2-3De v1.0)" in the Geoscientific Model Development (GMD) Journal**

Emmanuel Wyser, Yury Alkhimenkov, Michel Jaboyedoff, Yury Podladchikov

November 10, 2021

The referee comments appear in black, whereas our responses appear in blue and the changes made in the revised manuscript appear in red.

**1 Topical Editor**

Dear authors,

Thank you very much for this revised version of the manuscript, which addresses appropriately most comments of the reviewers. As for the remark about the length of the paper, I agree that the description of the material point method is helpful and justifies a slightly longer manuscript. In this regard, it is unfortunate that the revised version is even longer, yet I found the added content regarding the new multi-GPU implementation to be well worth the extra pages. Note however that the new content now requires some more peer-reviewing.

I would also like to point out, at the next opportunity, that any request/question/comment from the reviewers should be addressed in the manuscript, even though this can be done succinctly with more justifications or details provided in the "response to reviewers" documents. For instance, reply #1 for reviewer #1 should at least appear as a note. Similarly, reply #2 for reviewer #2 could probably be better integrated in the text as ("B-bar" without a reference, "closer" than what?...).

Best regards, Thomas Poulet.

**Reply** We would followed the topical editor's comment as much as possible. We addressed in the manuscript requests/questions/comments from the reviewers.

**2 Reviewer #1**

The authors implemented an explicit GPU-based solver within the material point method framework and tested using two- and three-dimensional problems. Results seem to agree with the expected values validating this MPM - GPU architecture. I would like to suggest the publication of this work. Nonetheless, some minor points should be addressed first.

We would like to acknowledge the reviewer for the time spent on the revision of our work.

**Comment # 1** The authors mentioned that this GPU architecture speeds up information transfer between nodes and material points. As stated, this is one of the most computationally expensive operations in MPM. Nevertheless, finding material points new location after the mesh returns to its original position is another process that is computationally expensive (in many cases more expensive than information transfer between

nodes and material points). I would like to know if the GPU architecture proposed also improves this step. If yes, the author could indicate it in the paper. If not, it would be interesting if the author discusses the possibility of combining some techniques (e.g. Pruijn N.S. 2016) together with GPU's to improve MPM computations.

Pruijn N.S. 2016. The improvement of the material point method by increasing efficiency and accuracy. TU Delft Master Thesis.

**Reply # 1**  The reviewer points out the computationally expansive operation of finding material point's new locations after the mesh has been reset at the end of a time step. We used a regular background mesh (as opposed to triangular mesh or non-constant element size), therefore, it is straightforward to find material point's new location. To find in which element $e$ a material point $p$ is located, we use the following equation

$$e = (\texttt{floor}((z_p - z_{\min})/\Delta z) + n_{el,z} \times \texttt{floor}((x_p - x_{\min})/\Delta x)) + n_{el,x} \times n_{el,z}\texttt{floor}((y_p - y_{\min})/\Delta y), \quad (1)$$

where $n_{el,x}$ and $n_{el,z}$ are the number of elements along $x-$ and $z-$directions, $x_{\min}$, $y_{\min}$ and $z_{\min}$ are the minimum $x$, $y$, and $z$ coordinate of nodes. However, this is far less trivial when irregular background mesh is used and such equation can not be used any more. Such concern is out of the scope of our contribution since our implementation only consider a regular background mesh. This concern also explain why we selected a regular background mesh.
  We looked at the reference suggested by the reviewer. From an overlook of the mentioned research work, we think a GPU-based implementation of the brute force method is possible, but this would require deeper investigations. This could be the subject of future studies. We strongly think such method could benefit from the computational power of GPUs.

**Change # 1**  P.10, L.235
We use regular background mesh because it is straightforward to find the material point's location. However, computing a material point's location using an irregular background mesh is more complicated.

**Comment # 2**  The authors include a damping value $D$ in the simulations. This value is not well validated. It seems that several simulations were needed to find it, giving the idea that $D$ is not related to the material properties and geometry and is more of an artificial way to reach the desirable results. The authors should elaborate better on the reasons for using this specific damping value.

**Reply # 2**  It is true that the damping value is not well validated. From a broader perspective and to our knowledge, no studies thoroughly investigated and quantified the influence of damping. However, the common range between 0.05 and 0.15 is usually selected by researchers (e.g., Wang et al., 2016b; Wang et al., 2016a) for dynamic analysis. This range was found sufficient to damp out dynamic oscillations while not producing spurious plastic yielding or an over-damped system, as mentioned in Wang et al., 2016a. As such, we decided to use a damping value of $D = 0.1$, since reasonable propagations were obtained and no spurious plastic yielding were noticed.
  As raised by the reviewer, we will elaborate better on the reasons of this specific value and will clarify accordingly the lack of validation of this damping value within the main body of the text. We will also mention the need for future studies addressing this concern.

**Change # 2**  P.30, L.581-585
Due to our explicit formulation, a damping relaxation term should be introduced to mitigate dynamic wave propagations (Wang et al., 2016b). In this work, we selected damping values that were either commonly accepted (e.g., $D = 0.1$ for slumps) or that were better resolving experimental results (e.g., $D = 0.025$ for granular collapses). Future investigations should specifically address the influence of damping terms on the material's behavior.

**Comment # 3**  In line 32, the authors mentioned that "The background mesh can be reset". As far as I know, the background mesh must be reset. I recommend changing the verb "can" for a better one.

**Reply # 3**  We agree with the reviewer. We will change the verb in the revised manuscript.

**Change # 3**  P.2, L.33
The background mesh is reset and actually never deforms.

**Comment # 4**  I am wondering if the variables in line 75 are the same as in equations 2 and 3 since different punctuations were used (e.g. $\AA_\ll$ and $\hat{u}$).

**Reply # 4**  We recognize here that this is a mathematical typo. Variables in line 75 are the same as in Eqs. 2 and 3. We will fix this in the revised version of the manuscript.

**Change # 4**  P.3, L.76
$\hat{u} \to \bar{u}$ and $\hat{\tau} \to \bar{\tau}$

**Comment # 5**  Finally, I recommend reading the paper again to correct some typos detected..

**Reply # 5**  We acknowledge the reviewer's recommendation and we will read the manuscript again to correct remaining typos

**Change # 5**  -

**3   Reviewer #2**

This paper demonstrates the numerical implementation of the explicit material point method using GPU for parallel computing. In Reviewer opinion, the paper is of interest for the Readers of Geoscientific Model Development. However, a number of issues need to be addressed before the paper can be accepted for publication.

We would like to acknowledge the reviewer for the time spent on the revision of our work.

**Comment # 1**  The paper is currently long and several part can be replaced by references. For example, the MPM algorithm is derived from forward-Euler scheme with update stress lass, GIMP basis functions in appendix A.

**Reply # 1**  Thank you for raising that point. We recognize that the paper can be long for the audience. However, we also think it is the bare minimum to have a grasp over the material point method. We strongly believe this avoids the reading of numerous references and facilitate the overall understanding of the method to a broader audience. We already attempted to shorten the section 2 by putting in Appendix a detailed description of the shape functions used in GIMPM.

**Change # 1**  -

**Comment # 2**  There are several methods dealing with the volumetric locking in the literature. However, the author proposed the volumetric locking by averaging only the volumetric part of the stress tensor. The author is suggested to clarify the decision for that. Furthermore, volumetric locking can smooth out the value of the stress. It is better if the stress is plotted in the numerical examples to see the difference between simulations with and without volumetric locking.

**Reply # 2**  It is true that in the literature, the stress tensor $\boldsymbol{\sigma}$ is averaged (Mast et al., 2012; Cuomo et al., 2019; Lei et al., 2020). Volumetric locking procedures generally mitigates locking by a more appropriate formulation of the volumetric component of the strain. The B-bar method ($\bar{B}$, see Bisht et al. 2021) splits the strain tensor into deviatoric and spherical parts. Deviatoric strains are evaluated at the material point's coordinate whereas the spherical part is evaluated only at the element center's coordinate. This yield a stress tensor, for which the pressure is defined at the element's center.

As such, we modified the element-based strategy (Mast et al., 2012; Cuomo et al., 2019; Lei et al., 2020), considering only the pressure term, which need to be averaged at the element's center. We believe this formulation is more consistent with the essence of the B-bar method, which splits the deviatoric part from the spherical part.

It is true that volumetric locking can smooth out the value of stresses. As suggested, we will add plots to show the difference between a locking-free and a locking-prone solution in the revised manuscript in the Appendix C: Volumetric locking and damping corrections.

We here show an example of slumping considering the geometry as in Model 2b. We selected an homogeneous initial cohesion field with values presented in the submitted manuscript. Figure 1 demonstrates that a significantly smoother pressure field is resolved with the proposed method. In addition, the pressure field is smoothed but it does not significantly differs from the original pressure field (in locations where locking is minimum). Volumetric locking is particularly highlighted within shear bands due to isochoring plastic flows, resulting in significant stress oscillations.

**Change # 2.1**  P.11, L.275-276
We believe our approach is conceptually similar to the B-bar technique (Hughes, 1980; Bisht et al., 2021).

**Change # 2.2**  P.33, L.670-676. Appendix C: Volumetric locking and damping corrections

We further present additional three-dimensional results for Model 2b for a homogeneous cohesion field (see Figs. 2 and 3). Three-dimensional simulations of cohesive material better illustrate the influence of volumetric locking. Figure 3 demonstrates that a significantly smoother pressure field is resolved with the proposed method.

In addition, the pressure field is certainly smoothed, but it does not significantly differs from the original pressure field (in locations where locking is minimum). Volumetric locking is particularly highlighted within shear bands due to isochoric plastic flows, resulting in significant stress oscillations.

[Figure]

Figure 2: Non-smooth pressure field due to volumetric locking. The typical check-board pattern of volumetric locking can be observed where the material has yielded, i.e., the shear band.

[Figure]

(a) Non-smooth pressure field due to volumetric locking

[Figure]

(b) Smooth pressure field when volumetric locking is mitigated with the proposed solution

Figure 1: Element-based reconstruction of the pressure field to mitigate volumetric locking issues, with a total number of material points $n_{mp} \approx 10^5$.

[Figure]

Figure 3: Smoother pressure field when volumetric locking is mitigated with the proposed solution.

**Comment # 3**  Section 4 mentions that Model 1b demonstrates the influence of mesh resolution but I do not see it in Model1b. The author is suggested to perform convergence rate analysis in different mesh size in the plane strain to highlight the influence of the mesh resolution.

**Reply # 3**  We understand the concern of the reviewer. It is true that the mesh resolution is not clearly demonstrated in Model1b under plane strain conditions.

We here present additional plots (see Fig. 5) to show to the reviewer the influence of the mesh resolution over shear banding. All the parameters used are the same as described in the original manuscript, except that the number of element in the $x$-direction is increased, *i.e.*, $n_{el,x} = 40, 80, 160, 320$.

[Figure]

Figure 4: Equivalent plastic strain for a variety of mesh resolution. One can see the influence of the mesh resolution over shear banding.

We can clearly observe that as the mesh resolution increases, the shear band resolution gets more accurate, *i.e.*, the finer the resolution the finer the shear band thickness. We also observe a more complex shear banding pattern at the bottom right, *e.g.*, $x > 140$ mm and $y < 50$ mm. Again, such shear banding arrangement is better resolved with a finer mesh resolution. We believe Fig. 5 clearly demonstrates the influence of the mesh resolution. Therefore, we do not think a convergence analysis is further needed.

**Change # 3**   P.17-18, L.377-380.

Furthermore, Figs. 5 a) and b) demonstrate the influence of the mesh resolution over shear banding: the finer the background mesh, the thiner the shear bands. This is significant since it shows that the dynamics of shallower granular avalanches appears more complex for higher resolutions.

[Figure]

Figure 5: (a) $\epsilon_{\text{eqv}}^p$ for the zoomed-in area in Fig. 9. A shallow granular flow clearly appears, as suggested by the higher values of $\epsilon_{\text{eqv}}^p$. This supports evidence of shallower granular avalanches during collapses. Deeper structures, which result in lower accumulated plastic strains, probably highlight slower deformation modes along well-defined and persistent shear bands. (b) $\epsilon_{\text{eqv}}^p$ for a coarser background mesh resolution, which demonstrates the influence of the mesh resolution over shear bands.

**Comment # 4**   For Model 1b, the presented final geometry of the experiment is shorter to the one in Bui et al. (2008) experiment (see Figure 6) in my opinion. Please check. Therefore, it is not necessary to introduce the damping.

**Reply # 4**   Thank you for this remark. We checked and the reviewer is correct. We performed additional simulations and still noticed that simulations without damping leaded to higher run-outs and softer response

of the material. We will clarify this in the revised version of the manuscript.

**Change # 4**  P.15, L.351 and P. 17, L.373. We corrected Figures 6 and 8.

[Figure]

Figure 6: Final geometry of the granular collapse for three-dimensional configuration of our GPU-based explicit GIMPM implementation ep3De v1.0. The green region (i.e., the intact region) is defined by the $L_2$-norm of the material point displacement $u_p = ||\boldsymbol{u}_p||_2 \leq 0.5$ mm, whereas the red region (i.e., the deformed region) is defined by $u_p = ||\boldsymbol{u}_p||_2 > 0.5$ mm. The experiment of Bui et al., 2008 is indicated by the blue dashed line (i.e., the free surface) and the blue dotted line (i.e., the failure surface).

[Figure]

Figure 7: Final geometry of the granular collapse for the plane-strain configuration for our GPU-based explicit GIMPM implementation ep2De v1.0. The numerical solution and the experimental results are in good agreement. Some differences are more pronounced when compared with the numerical results obtained under a three-dimensional configuration.

**Comment # 5**  In Model 2, there is a boundary effect on the failure mechanism as the shear band can touch the bottom boundary. It would be better if there is a larger depth in the bottom direction. And, the Model 2 introduces local damping which in my opinion it is not necessary.

**Reply # 5**  We acknowledge the boundary effect on the failure mechanism. However, our concern was to show an example of large deformation. A larger depth would also significantly reduces the total amount of deformation. Model 2 shows that several elasto-plastic features are resolved, such as bulging or thrusting at the toe of the slope. Such features would not be as obvious when a larger depth is considered. Concerning the introduction of a local damping, it is mainly to prevent an excessive run-out of the material when considering free-slip boundary conditions at the bottom of the computational domain.

However, we show here an example of a similar setting while considering a larger depth, as suggested by the reviewer. We briefly describe parameters that needed to be changed regarding the increase of the depth.

Regarding the initial geometry, we increase by a factor of two the $z$-direction and the $y$-direction. To avoid a significant elastic compaction of the material, we selected a higher Young's modulus, i.e., $E = 10$ MPa. This reduces the amount of vertical elastic compaction of the material during the elastic loading stage. No-slip boundary conditions are enforced at the base of the material. As thought by the reviewer, the local damping can be reduced. But a small value is still needed to suppress elastic wave propagation. In this case, we selected a damping value $D = 0.025$. An isotropic Gaussian random field is still selected for the cohesion field, with the same parameters used for Model 2b in the submitted manuscript.

Figure 8 shows the total displacement field after plastic yielding. In comparison with results in Model 2b, the magnitude of deformation is smaller. For instance, the intense bulging reported in the original manuscript is less evident in this setting. However, we still report thrusting mechanisms at the toe of the slope. Heterogeneous displacements are still observed, more evidently at the toe.

[Figure]

Figure 8: Total displacement after elasto-plastic loading.

We also report a principal shear band and a heterogeneous crown-like structure (see Fig. 9). Plastic strain localisation differs regarding what was reported in the original manuscript. However, other simulations revealed that multiple shear bands can initiate successively. In Fig. 10, we can observe both shallower and deeper shear bands.

However, the deeper shear band is more developed than the shallower one. Similarly, we also report a more significant heterogeneous displacement field and a deeper thrusting mechanism at the toe of the slope. This is because of the random generation of the cohesion field, which can results in weaker and/or stronger local cohesion values. This also demonstrates an important influence of the initial cohesion field over the elasto-plastic response of the material.

Still, we think that the original results presented in the manuscript better represent the different mechanisms during the elasto-plastic deformation of an unstable material. It is true that increasing the depth greatly reduce the influence over the shear band propagation. We will clarify and discuss these considerations in the revised manuscript. We will also notify the reader that the boundary influence the propagation of the shear band due to the shallower depth mentioned by the reviewer.

**Change # 5**   P.19, L.396-400.

We assume this setting features the principal first-order characteristics of a typical rotational earth slump (Varnes, 1958; Varnes, 1978), i.e., a complex zone of scarps (minor and major) delimiting a crown-like structure, followed by a transition (or depletion) zone in which the material flows homogeneously along internal shear zones due to severe plastic strain localizations and, finally, a compression (or accumulation) zone resulting in complex thrusting at the toe of the slump. Because of the nature of the boundary condition at the bottom of the material (i.e., free-slip), an additional horizontal sliding component is introduced within the rotational part of the displacement. This results in stronger deformations, which we want to highlight.

[Figure]

Figure 9: Equivalent plastic strain after elasto-plastic loading.

However, the bottom boundary condition influences the shear band propagation and the overall behaviour by introducing a stronger horizontal component in the motion.

[Figure]

[Figure]

Figure 10: Total displacement (up) and equivalent plastic strain (down) after elasto-plastic loading.

**Author comment # 6** We additionally extended the original single GPU code and implemented a multi-GPU version using the message passing interface standard. We provide the new section below, which then will be included (with some simplifications) into the manuscripts during the revision stage.

**The Multi-GPU Code Implementation**

**Introduction**

One of the major limitation of `ep2-3De v1.0` is the on-chip memory. We demonstrated that an implementation of the material point framework quickly reaches the hardware limit of GPUs, even on modern architectures. It is then essential to overcome this limit in order to resolve larger computational domain with a greater amount of material points.

[revised manuscript text omitted]

**Change # 6** We introduced in multiple places in the revised manuscript the multi-GPU implementation presented above. Track changes in the revised version should be obvious.